



# Sub-cloud Rain Evaporation in the North Atlantic Ocean

Mampi Sarkar[1], Adriana Bailey[1], Peter Blossey[2], Simon P. de Szoeke[3], David Noone[4],
Estefania Quinones Melendez[3], Mason Leandro[5], and Patrick Chuang[5]

[1]National Center for Atmospheric Research, Boulder, Colorado
[2]University of Washington
[3]Oregon State University
[4]University of Auckland
[5]University of California, Santa Cruz

**Correspondence:** Mampi Sarkar (mampi@ucar.edu)

**Abstract.** Sub-cloud rain evaporation in the trade wind region significantly influences boundary layer mass and energy budgets. Parameterizing it is, however, difficult due to the sparsity of well-resolved rain observations and the challenges of sampling short-lived marine cumulus clouds. In this study, rain evaporation is analyzed using a one-dimensional model that simulates both changes in drop size and changes in drop isotopic composition. The model is initialized with raindrop size distributions and water vapor isotope ratios (e.g. $\delta D$, $\delta^{18}O$) sampled by the NOAA P3 aircraft during the Atlantic Tradewind Ocean-Atmosphere Mesoscale Interaction Campaign (ATOMIC). Sensitivity tests suggest that the concentration of raindrops ($N_0$), the geometric mean diameter of the drops ($D_g$) and the width of the raindrop size distribution ($\sigma$) significantly control sub-cloud rain evaporation fluxes ($F_e$). While $N_0$ determines the overall magnitude of $F_e$, $D_g$ and $\sigma$ determine its vertical structure. Overall, the model suggests 65% of the rain events sampled at 1-Hz by the P3 over 22 in-cloud cases during ATOMIC evaporates into the sub-cloud layer. To assess the representativeness of these results, we leverage the fact that the percentage of rain that evaporates is proportional to the change in the deuterium excess (d=$\delta D$-8$\times\delta^{18}O$) of the drops between cloud base and the surface . We compare the deuterium excess simulated by the model with surface isotopic observations from the NOAA Research Vessel Ronald H. Brown. We find that the Brown must have sampled in conditions with higher surface relative humidity, larger cloud-base $D_g$, and larger cloud-base $\sigma$ than the P3. Overall, our analysis indicates that both thermodynamic and microphysical processes have an important influence on sub-cloud rain evaporation in the trade wind region.

## 1 Introduction

Shallow precipitation, a ubiquitous feature of marine cumulus clouds in the trade-wind tropical ocean basins, is significant in maintaining global moisture and energy budgets (Brost et al., 1982; Nicholls and Leighton, 1986; Paluch and Lenschow, 1991; Short and Nakamura, 2000; Jensen et al., 2000; Stevens, 2005). Rain rates on the scale of 1 mm/day, commonly associated with shallow cumulus precipitation, are capable of producing roughly 30 $Wm^{-2}$ of evaporative flux ($F_e$) in the sub-cloud layer (figure 1), which is comparable to the radiative and surface fluxes computed from mixed layer models (e.g. Caldwell et al., 2005) and within stratocumulus-to-cumulus transition regions (e.g. Kalmus et al., 2014). Such large $F_e$, although localized, can create stability between cloud and sub-cloud layers, feed cool and moist air into large-scale circulations (Stevens, 2005),



form cold pools (Jensen et al., 2000; Zuidema et al., 2012; de Szoeke et al., 2017), and facilitate cloud break-up (Paluch

and Lenschow, 1991; Sandu and Stevens, 2011; Yamaguchi et al., 2017; O et al., 2018; Sarkar et al., 2020). Thus, accurate estimation of the rain evaporation flux in shallow cumulus regions and characterization of its vertical structure are needed.

Past field campaigns sampling precipitation events over both the Atlantic and Pacific Oceans suggest substantial $F_e$ is produced by shallow cumulus systems. For example, during the Atlantic Stratocumulus Transition Experiment (ASTEX) conducted in June 1992 over the east-central Atlantic Ocean, stratocumulus rain events evaporated into the dry and deep sub-cloud

layer before reaching the surface (Bretherton and Pincus, 1995). A rough estimate from figure 6 in Bretherton and Pincus (1995) suggests that mean rain rates within cloud layers and near the surface were 2.4 mm/day and 1 mm/day, respectively, which is consistent with an $F_e$ of 42 $Wm^{-2}$.

In 2012-2013, the Rain in Cumulus over the Ocean (RICO) campaign off the Caribbean islands of Antigua and Barbuda was conducted. All flights except one on 19 January were designed to randomly sample clouds above cloud base, resulting in

most flights not sampling any precipitation. But the precipitation sampled on 19 January suggests a 6% reduction of rain rate below cloud base (Snodgrass et al., 2009), which roughly translates to 130 $Wm^{-2}$ of rain evaporation flux (based on figure 10 in Snodgrass et al. (2009)).

Similarly, the Cloud System Evolution in the Trades (CSET) mission was conducted in 2015 over the trade-wind north Pacific Ocean, where precipitation was extensively sampled within both stratocumulus and cumulus clouds between California

and Hawaii. Rain rates during CSET were much higher than rain rates sampled in similar campaigns over the Atlantic Ocean, frequently exceeding 1 mm/hr while also producing rain that reached the surface (Albrecht et al., 2019; Mohrmann et al., 2019; Sarkar et al., 2020). The $F_e$ during CSET increased from the stratocumulus to cumulus regions and reached 10-200 $Wm^{-2}$ within some heavily precipitating cumulus transects.

These previous studies support the idea that shallow precipitation is important to the local energy budget. However, more

rain observations are needed to characterize $F_e$ in trade wind regions in a statistically robust way. Moreover, questions remain about $F_e$ variability in different cloud conditions and its sensitivity to boundary layer microphysical and thermodynamic characteristics. Does the vertical structure of $F_e$ depend on thermodynamic changes in the sub-cloud layer or on microphysical processes? Does $F_e$ facilitate or hinder boundary layer stability at local scales? How sensitive is $F_e$ to microphysical processes like collision-coalescence above cloud base? In this study we attempt to characterize and quantify $F_e$, its vertical structure,

and its dependence on microphysical (i.e. raindrop concentration, size and distribution width) and thermodynamic (i.e. surface relative humidity) features using a one-dimensional steady-state evaporation model initialized by in-situ field observations.

The major challenges in constraining $F_e$ with observations are the difficulty of sampling rain in cumulus clouds, due to the temporal and spatial variability of precipitation, and the limitations in existing rain retrieval methods. The airborne mm-wavelength radar used during field campaigns provides a wide and homogeneous array of cloud and precipitation samples in

terms of radar moments. However, accurate microphysical retrievals from the radar moments are difficult due to Mie scattering and atmospheric and liquid attenuations (Fairall et al., 2018; Schwartz et al., 2019; Sarkar et al., 2021).

In comparison, in-situ cloud and rain probes, although limited in their sampling volume compared to radar, provide well-resolved, direct and accurate microphysical raindrop size distributions (RSD). In-situ measurements also provide stable isotope



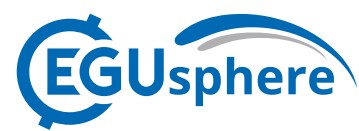

ratios of hydrogen and oxygen in water vapor, which can be used to independently assess rain evaporation. This is because
as rain evaporates into the unsaturated sub-cloud layer, fractionation leads to its isotopic enrichment as a function of relative
humidity and RSD at cloud base (Salamalikis et al., 2016; Graf et al., 2019).

This study makes a novel attempt to evaluate rain evaporation using both RSDs and water vapor isotope ratios measured in
situ during ATOMIC, the Atlantic Tradewind Ocean-Atmosphere Mesoscale Interaction Campaign, which was a component of
the international field campaign known as EUREC[4]A (ElUcidating the RolE of Clouds-Circulation Coupling in ClimAte). For
the first time, the isotopic enrichment of rain is modeled using RSDs measured in the field and evaluated using surface-based
isotopic rain observations.

## 2   Data and Methodology

### 2.1   EUREC[4]A/ATOMIC campaign and datasets

The Atlantic Tradewind Ocean-Atmosphere Mesoscale Interaction Campaign (ATOMIC) was conducted in the north Atlantic
trade wind region roughly within 51°W-60°W and 10°N-15°N to study mesoscale circulations in the atmosphere and ocean
(Pincus et al. 2021). ATOMIC was the NOAA-sponsored component of the larger international field campaign EUREC[4]A, or
ElUcidating the RolE of Clouds-Circulation Coupling in ClimAte, which took place in January-February 2020 near Barbados
(Stevens et al., 2021). Both the NOAA WP-3D Orion (P3) aircraft and the NOAA R/V Ronald H. Brown (Ron Brown) were
deployed as part of ATOMIC.

The P3, integrated with radar, in-situ instruments and dropsondes, was flown through cloud and rain transects to collect
thermodynamic and microphysical boundary layer observations and facilitate investigations of aerosol-cloud-precipitation in-
teractions. An example of the trajectory of the P3 is shown in figure 2 for 9 February, where a series of stacked 10-minute
horizontal legs were flown to sample the boundary layer extensively. The horizontal legs were flown at 150 m, 500 m, 700 m, 2
km and 3 km between 54°W-56°W. During most flights a level circle was also conducted at 7.5 km altitude, and 7-8 dropson-
des were released to obtain high-resolution thermodynamic observations reaching the surface. The surface relative humidity,
which is integral to the rain evaporation model used in this paper, is obtained at ≈10 m altitude. Its values for 9 Feburary are
noted at the dropsonde locations in figure 2b.

This paper characterizes rain structure during ATOMIC using the Cloud Imaging Probe (CIP) and Precipitation Imaging
Probe (PIP) instruments on board the P3, which sampled raindrop size distributions in in-situ mode. The CIP samples cloud-
and rain- drops across 25 $\mu$m-1.6 mm, while the PIP samples across 100$\mu$m- 6.2 mm (Pincus et al., 2021; Leandro and Chuang,
2021). The CIP and PIP observations are stitched together to obtain 1-Hz raindrop size distributions for diameters spanning
$\mu$m-6 mm (total 23 bins). Bin sizes smaller than 100 $\mu$m and bigger than 6 mm are not reliable and are not used in the
current analysis. Drops across 400-1800 $\mu$m, 1.8-5 mm, and 5-6 mm drop sizes are binned at 200 $\mu$m, 400 $\mu$m, and 1 mm
resolutions, respectively.

This paper is based on the later part of the ATOMIC campaign when the CIP and PIP instruments were properly functioning
and mean rain rates greater than 0.01 mm/day were observed during 22 10-minute horizontal legs on 4, 5, 9 and 10 February.





The rain rates are calculated from the observed RSD using $R = 6\pi \times 10^{-7} \sum_{i=125\mu m}^{6mm} N_i v_i D_i^3$, where $N_i$ is the raindrop concentration for drop diameter $D_i$, $v_i$ is the terminal velocity associated with $D_i$, and i is the index of the RSD bin. The rain rates for each 10-minute leg are averaged and noted in the legend of figure 3. Even though the probe instruments were working on

31 January and 11 February, the mean rain rates from those days were below 0.01 mm/day during all of the 10-minute in-cloud horizontal legs and are therefore not included in this study.

To model the isotopic evolution of the RSDs, 1-Hz water vapor mixing ratio and water vapor isotope ratios for hydrogen and oxygen ($\delta D_v$ and $\delta^{18}O_v$ respectively) were obtained from the Picarro L2130-i water vapor isotope analyzer flown during ATOMIC (Bailey et al., 2022). When airborne, the analyzer drew in ambient air through a 0.25 inch backward-facing tube,

which ensured the selective sampling of water vapor as opposed to liquid water (Pincus et al., 2021). $\delta D_v$ and $\delta^{18}O_v$ represent the ratios D/H and $^{18}O/^{16}O$, respectively, normalized to VSMOW (Vienna Standard Ocean Water) and reported in units permil (‰). The isotopic enrichment of rain in the sub-cloud layer due to rain evaporation is evaluated in this study using precipitation isotope ratio observations made on the NOAA Research Vessel Ronald H. Brown (ship). During ATOMIC, the Brown sailed in the trade-wind region between Barbados and the Northwest Tropical Atlantic Station (NTAS), a buoy-station near 15°N,

51°W, to provide a ground-based perspective for the P3 flying overhead. The Brown collected 12 samples across a wide geographic area between 5 January and 11 February (Bailey et al., 2022, submitted). Additionally, rain isotope ratios and rain rate measurements from the ship are used to evaluate the accuracy of the sub-cloud rain evaporation model used in this study. While the Brown measurements are not precisely co-located in space or time with the P3, they still provide a useful assessment of the trade-cumulus environment in which the P3 flew.

Isotope ratios were also sampled in surface precipitation at two other stations, viz. the Barbados Cloud Observatory (BCO) and the German R/V Meteor (ship), that were part of the EUREC⁴A campaign. These provide further observational constraints for the rain evaporation model used in this study. The BCO is a land-based observatory on the eastern shores of Barbados, where 42 precipitation samples were collected from 16 January and 18 February (Villiger, 2021). The Meteor sailed along a north-south transect defined by the 57.24◦W meridian and sampled 15 rain events between 20 January and 19 February.

Uncertainties associated with the Meteor samples are 0.2‰ and 0.5‰ for $\delta^{18}O_p$ and $\delta D_p$, respectively (Galewsky, 2020). More details regarding isotopic observations, stations and measurement techniques during ERUEC4A are described in (Bailey et al., 2022, submitted).

## 2.2   Sub-cloud rain evaporation model

Observed raindrop size distributions are used to initialize the sub-cloud rain evaporation model on 4, 5, 9 and 10 February.

This one-dimensional steady-state model is used mainly to (a) estimate the amount and vertical distribution of water vapor and the equivalent latent flux produced by the evaporation of raindrops of 125 $\mu$m-6 mm diameters, and to (b) estimate the change in precipitation isotope ratios $\delta D_p$ and $\delta^{18}O_p$ of the raindrops during evaporation. The model follows the numerical isotope-evaporation model described in detail in Graf et al. (2019) and Salamalikis et al. (2016) and incorporates vertical variations in drop size and drop temperature. Any collision-coalescence process for raindrop growth and other raindrop-cloud droplet interactions are ignored. Vapor contributed by rain evaporation is also neglected by this steady-state model.





Cloud base in the model is deduced from the ceilometer observations from the Brown. The 10-minute resolved ceilometer observations (Quinn et al., 2021) show that the median cloud bases on 4, 9 and 10 February are within 700-800 m. Consequently, the raindrops are initiated from a 700 m cloud base and modeled to fall through a sub-saturated sub-cloud layer. The relative humidity at cloud base is assumed to be 100%, decreasing linearly towards the surface (verified from the dropsonde

observations) with surface relative humidity varying from 65%-80%, as determined from nearby dropsonde observations. The rain water content (RWC) is computed at cloud base using the stitched CIP/PIP-based raindrop size distribution.

The change in raindrop size is calculated every 50 m as raindrops fall from the cloud base using:

$$\frac{dD}{dz} = \frac{4(F_v D_{va})}{Dv\rho_w R_v}[RH\frac{e_{v,sat}}{T_a} - \frac{e_{r,sat}}{T_r}] \tag{1}$$

where $F_v$ (unitless), $D_{va}$ (m$^2$s$^{-1}$), $R_v$ (461.5 Jkg$^{-1}$K$^{-1}$), RH (%), $e_{v,sat}$ (Pa), $e_{r,sat}$ (Pa), $T_a$ (K), $T_r$ (K), D (m), v (m/s),

$\rho_w$ (10$^6$ g/m$^3$) and z (m) are the mass ventilation coefficient, diffusivity of water vapor in air, gas constant for water vapor, relative humidity, saturation vapor pressure at ambient temperature and drop surface, ambient temperature and raindrop surface temperature, raindrop diameter, raindrop terminal velocity and vertical height-level, respectively. The vertical variation of $T_r$ is given by:

$$\frac{dT_r}{dz} = \frac{12F_h k_a}{D^2 \rho_w c_w v}((RH\frac{e_{v,sat}}{T_a} - \frac{e_{r,sat}}{T_r})(\frac{F_v D_{va} L}{F_h k_a R_v}) + (T_r - T_a)) \tag{2}$$

where $F_h$ (unitless), $k_a$ (Jm$^{-1}$s$^{-1}$K$^{-1}$), $c_w$ (Jg$^{-1}$K$^{-1}$) and L (J/g) are the heat ventilation coefficient, thermal conductivity of air, water density, specific heat of water and latent heat of vaporization, respectively.

The calculated D at vertical level z is used to model RWC (gm$^{-3}$) using:

$$RWC = \frac{\pi}{6}\rho_w \sum_{i=1}^{23} N_i D_i^3 \tag{3}$$

where $N_i$ is the concentration of raindrops at each diameter bin $D_i$, i being the bin index. The total water vapor produced from

raindrop evaporation within a 50 m vertical grid box is obtained from the difference of RWC at every 50 m vertical level, i.e. $q_e(z)(gm^{-3}) = RWC(z) - RWC(z-50)$. The total evaporated moisture content produced in the sub-cloud layer of 700 m altitude is calculated using the difference between RWC at 700m and surface ($q_{eT}(gm^{-3}) = RWC(700m) - RWC(surface)$).

Similarly, the precipitation flux $F_p(z)$ (Wm$^{-2}$) at each level z is modeled using:

$$F_p(z) = \frac{\pi}{6}\rho_w L \sum_{i=1}^{23} v_i N_i D_i^3 \tag{4}$$

. The rain evaporation source $S_e(z)$ (Wm$^{-3}$) at level z is then calculated using the difference between $F_p$ at level z and z-50 m:

$$S_e(z) = -\frac{\delta F_p(z)}{\delta z} = \frac{F_p(z) - F_p(z-50)}{50}. \tag{5}$$

The total rain evaporation flux produced over the entire sub-cloud layer $F_{eT}$ (Wm$^{-2}$) is obtained from:

$$F_{eT} = \int_{surface}^{700m} S_e(z)\,dz = F_p(700m) - F_p(surface). \tag{6}$$





Similarly, the rain evaporation flux between two consecutive layers is given by $F_e(z)$ (Wm$^{-2}$):

$$F_e(z) = S_e(z)\Delta z = F_p(z) - F_p(z - 50).$$ (7)

The vertical change in $\delta_p$ (‰) of raindrops as they evaporate is given by,

$$\frac{d\delta_p}{dz} = \frac{12 e_{r,sat} F_v D_{va}}{\rho_w R_v D^2 T_r v} \times [(\frac{D_{va}^i}{D_{va}})^n ((\delta_{va} + 10^3) RH \frac{e_{v,sat} T_r}{e_{r,sat} T_a} - (\frac{(\delta_p + 10^3)}{\alpha_{p \to v}})) - (\delta_p + 10^3)(RH \frac{e_{v,sat} T_r}{e_{r,sat} T_a} - 1)]$$ (8)

where $\delta_p$ applies to both $\delta D_p$ and $\delta^{18}O_p$ with n=0.58. Equation 8 includes the effects of both evaporation of raindrops and the
exchange of isotopes between raindrops and ambient vapor during equilibration. $\delta_{va}$ is the ambient water vapor isotope ratio
expressed in ‰, obtained from in-situ mean 150 m level isotope ratio observations. All the parameters in equations 1-8 are
obtained from Graf (2017).

The rain isotope ratios used to initialize $\delta_p$ at cloud base are determined using in-situ water vapor isotope ratios and the
measured temperature by assuming that the raindrops are in equilibrium with the water vapor (Risi et al., 2020) and scaling by
a temperature dependent equilibrium fractionation factor $\alpha_{p-v}$ (=($\delta_p$/1000 +1) /($\delta_v$/1000 +1)), as defined in Majoube (1971).
The modeled $\delta_p$ at the surface is later compared with the surface rain isotope ratio observations from the BCO, the Brown and
the Meteor to validate the accuracy of the model.

Evaporation leads to an increase in the $\delta D_p$ and $\delta^{18}O_p$ of the falling raindrops in the sub-cloud region, where small drops
enrich more than larger drops. The changes in $\delta D_p$ and $\delta^{18}O_p$ are not calculated once the raindrops evaporate below 350 $\mu m$
diameter. The modeled isotope ratios for drops larger than 500 $\mu m$ are consistent for vertical grid resolution of 1 m and 50 m
(11), giving us confidence in using the model at 50 m vertical resolution.

When the rain evaporated water vapor mixes with the ambient water vapor then the total water vapor at a given vertical level
is given by:

$$q_v = q_e + q_{va}$$ (9)

where $q_v$, $q_e$ and $q_{va}$ are the absolute humidity (in gm$^{-3}$) of the total, rain evaporated and ambient water vapor. $q_v$ and $q_e$ are
calculated at every 50 m altitude. $q_{va}$ is assumed to be constant within the sub-cloud layer and the effect of rain evaporation
on $q_{va}$ is considered negligible.

Similarly, the net isotope ratio at a vertical level z is a combination of the background vapor isotope ratio and the isotope
ratio evaporated from the drop, given by Noone (2012):

$$\delta_v = \frac{\delta_e q_e + \delta_{va} q_{va}}{q_e + q_{va}}$$ (10)

where $\delta_e$, $\delta_{va}$ and $\delta_v$ are the isotope ratios (‰) of the evaporated rainwater, ambient water vapor and total water vapor at level
z, respectively. $\delta_e(z)$ is computed from the difference between the product of $\delta_p$ and RWC at every 50 m level i.e.

$$\delta_e(z) = \frac{\delta_p(z)RWC(z) - \delta_p(z - 50)RWC(z - 50)}{q_e(z)}$$ (11)





Vertical variations in $\delta_v$ are modeled too in order to study the effect of rain evaporation on the total water vapor isotope ratio,
and whether or not the effect can be resolved by the current generation of flight-ready, real-time water vapor isotopic analyzers.

The role of microphysical processes in influencing modeled rain evaporation is investigated in terms of the total raindrop concentration ($N_0$), geometrical mean diameter ($D_g$) and the lognormal distribution width ($\sigma$) at the sampling level. These parameters ($N_0$, $D_g$ and $\sigma$) provide physically meaningful quantities to interpret microphysical conditions of rain, are helpful in evaluating the sensitivity of rain evaporation to microphysical changes, and are derived by fitting the observed RSDs to a
lognormal distribution following:

$$N(D) = \frac{N_0}{D\sqrt{2\pi ln^2\sigma}}exp(\frac{-(lnD-\mu)^2}{2ln^2\sigma}) \tag{12}$$

where $\mu$ is the log of $D_g$ ($D_g = e^\mu$). $N(D)$ is substituted into equation 3-8 to obtain RWC, $F_p$ and $F_e$.

## 3 Results and discussion

### 3.1 Rain characteristics

Rain rates sampled during 20 out of the 22 10-minute (1 Hz) horizontal flight legs on 4, 5, 9 and 10 February (figure 3) are low with leg-mean rain rates varying within 0.01-3 mm/day and rain frequency within 1-10%. Somewhat more intense rain was sampled during two cases on 9 February, measured at 1630 m and 2112 m, with rain rates of 22 mm/day and 31 mm/day and rain frequencies of 10% and 50%, respectively. The leg-mean rain rates are calculated over the raining samples (rain rate>0.01 mm/day) only and the rain frequency is defined as the ratio of the number of raining samples to the total number of raining and
non-raining samples within a 10-minute horizontal leg. Seven out of the 22 cases were sampled within ±100 m of cloud base (700 m) compared to the other cases where sampling altitudes were generally higher than 1.3 km. The highest and lowest rain rates were observed on 9 February (31 mm/day) and 4 February (0.01 mm/day), respectively. The low rain rates on 4 February are due to the higher concentration of small raindrop diameters (<200 $\mu$m) compared to the other days. Comparatively higher rain rates at 1630 m and 2112 m altitudes on 9 February are due to the much higher raindrop concentrations and bigger drop
sizes. No rain was detected at any of the 150-m altitude legs (except one case on 9 February), suggesting that rain either evaporated completely before reaching the surface or was not sampled when it reached the surface.

Vertical and horizontal variability in rain structure is evident in the radar images (e.g. figure 4), which reveal heterogeneous cloud bases and some heavy precipitation pockets with radar reflectivity higher than -10 dBZ. These heavier precipitating samples partially evaporate before reaching the surface. The more weakly precipitating segments, with smaller radar reflectivities,
evaporate completely within the sub-cloud layer. Vertical changes in rain structure are also evident from in-situ observed RSDs measured at different altitudes within the same cloud system. For example, on 9 February the RSDs shift towards smaller drop sizes as sampling altitudes decrease from 2112 m to 1630 m to 1500 m to 1053 m (figure 3c), which could be due to both microphysical and thermodynamic processes in the cloud and sub-cloud layers.



## 3.2   Observed microphysical and thermodynamic variability

A strong positive correlation is observed between the rain rates and microphysical parameters $N_0$, $D_g$ and $\sigma$ at cloud base
(figure 5a-c). The higher $D_g$ and $\sigma$ indicate higher concentration of larger drops, which account for more liquid water and the
higher rain rate. While $D_g$ does not vary a lot (0.16-0.26 mm), $N_0$ and $\sigma$ vary within $10^2$-$10^3 m^{-3}$ and 1-3, respectively, over
the 22 P3 cases. The corresponding rain rates are between 0.01-35 mm/day.

Comparing the 22 ATOMIC cases with five cases of raining cumulus clouds from the CSET campaign, which were measured
over the Pacific Ocean, we find that CSET was characterized by higher rain rates (1-100 mm/day), along with higher $N_0$ ($10^3$-
$2\times10^4 m^{-3}$) (figure 5, Sarkar et al. (2020)). However, the average $D_g$ and $\sigma$ from the five CSET cases (0.2 mm and 2.3,
respectively) are within the P3 $D_g$ and $\sigma$ ranges. Since $D_g$ and $\sigma$ did not vary significantly across the five CSET cases, the
higher rain rates during CSET were due to the two orders of higher magnitudes of $N_0$.

The 0.01-5 mm/day rain rates sampled by P3 (figure 5-6) are also much lower compared to the 5 mm/hr cloud base rain rates
sampled during RICO (figure 2, Geoffroy et al. (2014)). The 1-Hz distribution of $N_0$, $D_g$, $\sigma$ and rain rates are plotted in figure
6 for an overall microphysical statistical characterization for all the P3 cases. The variability in the rain parameters is the least
on 4 February and highest on 9 February. Comparing figure 6 with figure 2 in Geoffroy et al. (2014), the median rain rates, $N_0$
and raindrop diameters during RICO at cloud base were much higher than ATOMIC. Geoffroy et al. (2014) have used mean
volume diameter $D_v$ to describe the variability of their raindrop sizes, which is mathematically different but still comparable to
230   $D_g$ that we have used in this study. $D_v$ at 500 m during RICO is 750 $\mu$m which is much higher than $D_g$ of 200 $\mu$m during the
P3 cases. Similarly, the median $N_0$ and rain rates during RICO are $6\times10^4 m^{-3}$ and 12 mm/hr at 500 m altitude compared to
$300 m^{-3}$ and 0.4 mm/day respectively sampled on the highest raining case on 9 February by P3. This suggests that the higher
rain rates sampled by RICO were due to the high $N_0$ and $D_v$ clouds compared to the shallow precipitation by P3.

The ATOMIC cases also show a weak negative correlation between surface relative humidity ($RH_{sf}$) and rain rate (figure
235   5d). This may be due to the downdrafts drying the surface layer and lowering $RH_{sf}$. The $RH_{sf}$ values for 20 out of the 22 cases
are within 65-80%, which is smaller than during CSET, where 4 out of 5 cases were at 84%. The correlation between $RH_{sf}$
and rain rate during CSET was also weaker. That said, it is worth noting that $RH_{sf}$ measurements during CSET were collected
using aircraft observations at 150 m altitude and, therefore, might be slightly off from the actual surface relative humidity.

One assumption made in this study is that the RSD observed from the ATOMIC P3 accurately represents cloud base, even
240   when the sampling altitude was higher. Indeed, sampling altitudes were higher than 1300 m for 15 out of 22 cases, while the
ceilometer-based median cloud base during ATOMIC was just 700 m. The 7 cases where sampling altitudes were within $\pm100$
m of cloud base are shown in figure 5. Although the $N_0$ is lesser and $D_g$ is larger for these 7 cases than the other 15 cases,
the variation is in the order of just 10 $m^{-3}$ and 0.04 mm, suggesting that our assumption is appropriate. Slightly lesser $N_0$ and
larger $D_g$ could be due to collision-coalescence as raindrops fall towards cloud base.





### 3.3 Modeled rain evaporation in the sub-cloud layer

#### 3.3.1 Microphysical influences

The variability in rain rates calculated from the observed RSDs, over the 22 in-cloud legs is reflected in the modeled rain rates in the sub-cloud layer. The vertical profiles of modeled RWC, $q_e$ and $F_e$ are shown for all 22 cases in figures A1-A3. The highest modeled rain rates and RWC are on 9 February, where rain in 4 out of 5 cases reaches the surface, implying partial evaporation in the sub-cloud layer. In total, 11 out of 22 cases show complete evaporation, especially on 4 February, where, owing to RSDs centered at smaller drop sizes, rain completely evaporates within 350 m of cloud base.

The modeled microphysical parameters $N_0$, $D_g$ and $\sigma$ are shown in figure 7a-c for the four cases on 9 February (1630m, 2112m, 1500m, 2055m) where rain reaches the surface. The variation of $N_0$, $D_g$ and $\sigma$ can be explained in terms of the RSD at cloud base undergoing size sorting processes due to evaporation. As the rain evaporates below cloud base, the smaller drops evaporate faster than the larger drops. This can lead to the complete evaporation of smaller drops while larger drops partially evaporate, shrinking in size. Alternatively, raindrops of different sizes fall at different speeds leading to larger drops reaching closer to the surface before smaller drops. Both these processes may lead to a shift of the RSD towards larger $D_g$, narrower $\sigma$ and lower $N_0$.

The four cases on 9 February where rain reaches the surface have rain rates higher than 3 mm/day at cloud base (figure 7d). In particular, the 1630 m and 2112 m cases, with the highest rain rates at cloud base (>20 mm/day), also have the highest rain rates reaching the surface (>1 mm/day). The modeled rain rates are very sensitive to the decreasing $N_0$ in the sub-cloud layer as compared to changes in $D_g$ and $\sigma$. This is reasonable since rain rates are linearly proportional to $N_0$ and also because $N_0$ decreases much more rapidly toward the surface compared to $D_g$ or $\sigma$. For example, in the 2112 m case, the initial $N_0$ at cloud base is 1000 m$^{-3}$, which decreases to 100 m$^{-3}$ at the surface. In contrast, $D_g$ and $\sigma$ vary only within 0.2-0.75 mm and 1.7-3.2, respectively, over the entire sub-cloud layer. The larger sensitivity to $N_0$ compared to $D_g$ and $\sigma$ causes the rain rate to be dominated by changes in $N_0$. Therefore, the higher the $N_0$ at cloud base, the higher the rain rate.

The rain evaporation flux $F_e$ (figure 7e) is sensitive to both the effects of $N_0$ and the size sorting that takes place as drops evaporate. This is because when $F_e$ is expanded in terms of a lognormal distribution (substituting equation 11 into 3-5), one can see that it is directly proportional to both $N_0$ and the change in normalized RSD due to size sorting. Essentially, the higher the $\sigma$ at cloud base, the broader the RSD, and the more likely that larger raindrops reach the surface, making the $F_e$ profile more bottom heavy (assuming the same $D_g$ at cloud base). The $F_e$ profiles for the 2055 m and 2112 cases are both bottom heavy, with $\sigma$ greater than 2.5. However, since $N_0$ for the 2112 m case is much higher than for the 2055 m case, the magnitude of $F_e$ for the 2112 m case is also much higher.

How bottom- or top-heavy a profile of $F_e$ is, may have an effect on the boundary layer stability. For example, the generation of maximum moisture close to the cloud base (top-heavy) due to rain evaporation destabilizes and leads to top-down mixing of the sub-cloud layer, linking surface moisture to the cloud layer and helping the cloud layer stay intact. In contrast, a bottom-heavy $F_e$ profile stabilizes and inhibits mixing in the sub-cloud layer, reducing surface moisture flux to the cloud layer and thereby facilitating boundary layer decoupling. The two cases with higher $\sigma$ at cloud base in figure 7a also have bottom-heavy




$F_e$ profiles and vice-versa. This suggests that a higher $\sigma$ at cloud base is more likely to produce a bottom-heavy $F_e$ profile, and
hence a more stable boundary layer compared to a smaller $\sigma$.

The evaporated fraction of rain, which is computed as $1-R_z/R_{cb}$, where $R_z$ and $R_{cb}$ are the rain rates at altitude z and at cloud base, respectively, is also strongly sensitive to $D_g$ and $\sigma$. This is because the effect of $N_0$ nearly cancels out when normalizing the rain evaporated with the rain at cloud base. All told, 70-95% of rain evaporates before reaching the surface for the four 9 February cases. The two smallest $\sigma$ cases at cloud base correspond with evaporated fractions of 95% (figure 7f).

To further evaluate the importance of each microphysical parameter ($N_0$, $D_g$ and $\sigma$) on $F_e$ independently, we conduct a series of sensitivity tests (figure 8b-d). These show that for constant $D_g$ and $\sigma$, the higher the $N_0$ at cloud base, the higher the magnitude of $F_e$, without any effect on $F_e$'s vertical structure (figure 8b). However, for constant $N_0$, if $D_g$ and $\sigma$ increase, then the magnitude of $F_e$ increases and the maximum in the $F_e$ profile shifts towards the surface (figure 8c-d). In other words, the higher the cloud base $D_g$ and $\sigma$ are, the more bottom heavy the $F_e$ profile gets. This is reasonable because for a constant $N_0$, if $D_g$ or $\sigma$ increase, then the proportion of bigger drops increases too, which leads to a higher chance of bigger drops reaching the surface and a higher amount of liquid water available to evaporate. Overall, the distinct roles of $N_0$, $D_g$ and $\sigma$ in controlling the rain rate, the $F_e$ magnitude and profile, and the evaporated fraction imply that microphysical parameters influence key features of rain evaporation in the sub-cloud layer.

### 3.3.2 Thermodynamic influences

$RH_{sf}$ for the four cases on 9 February vary within 67-73%. According to our sensitivity analysis, this range is too narrow to cause any significant change in the modeled $F_e$ and rain rates. Higher variations in $RH_{sf}$, such as 50-90%, do affect $F_e$ magnitude and vertical structure (figure 11a). Depending on whether the background sub-cloud air that the rain falls into, originated from a saturated or unsaturated downdraft, or if the air is entrained from the free troposphere, the $RH_{sf}$ can be high or low. For constant $N_0$, $D_g$ and $\sigma$, a lower $RH_{sf}$ leads to greater and more top-heavy $F_e$ profile. A decrease in $RH_{sf}$ from 80% to 70% leads to an increase in $F_e$ from 107 to 120 Wm$^{-2}$. Similarly for a decrease in $RH_{sf}$ from 80% to 60%, $F_e$ increases from 107 to 127 Wm$^{-2}$ (legends in figure 11a).

The change in RSD after reducing $RH_{sf}$ (from 80% to 70% to 60%) is shown in figure 9. Even though the RSD at cloud base (700m) is the same for all cases, the surface RSD varies depending on the $RH_{sf}$. For 70% surface humidity (generally observed by the P3 dropsondes), all drops smaller than 900 $\mu$m that start falling from cloud base evaporate completely before reaching the surface. The larger the drop's size, the slower its diameter decreases ($\frac{dD}{dt} \propto \frac{1}{D}$). Drops 2-4 mm at cloud base evaporate negligibly before reaching the surface, whereas, drops of 1.1 mm diameter evaporate to 600$\mu$m at the surface.

At drier conditions (60% surface relative humidity), evaporation is slightly higher: 1.1 mm drops evaporate to 400 $\mu$m at the surface. In contrast, moister conditions (80% $RH_{sf}$) cause less evaporation, with complete evaporation occurring only for drops smaller than 700$\mu$m. In summary, the lower the $RH_{sf}$, the drier the sub-cloud layer and the higher the intensity of rain water evaporation, with the majority of evaporation taking place close to the cloud base. Changes in $RH_{sf}$ can affect both the magnitude and vertical structure of $F_e$ significantly, without any necessary changes in $N_0$, $D_g$ and $\sigma$.





### 3.3.3 Modeled rain flux at cloud base vs surface

The rain flux at cloud base (subscript: cb) $F_{p,cb}$ is computed from the RSD observations (equation 4) for all 22 cases. We compare this quantity to the cumulative rain evaporation flux $F_{eT}$ over the entire sub-cloud layer (equation 6). A scatter plot

between $F_{p,cb}$ and $F_e$ shows examples of complete and partial evaporation occurring over 20 of the 22 cases (figure 10). Two of the cases with the highest $F_{cb}$ and $F_e$ (on 9 February) are not shown in the plot for clarity. The cases where rain evaporates completely within the sub-cloud layer fall on the one-to-one line. The more the cases diverge from the one-to-one line, the higher the fraction of rain that reaches the surface. In general, the higher the $F_{p,cb}$ at cloud base, the higher the chance of rain reaching the surface. We suspect this is due to higher values of $D_g$ or $\sigma$ at cloud base, since RSDs with large $D_g$ and $\sigma$

have a greater chance of producing bottom-heavy $F_e$ profiles. The cases where samples were obtained closer to cloud base (700±100m) mostly show complete evaporation.

The slope of $F_{p,cb}$ versus $F_e$ in figure 10 is 0.65, denoting that on an average 65% of the rain evaporates before reaching the surface. Moreover, while $F_e$ varies greatly over the P3 cases, there are at least 8 out of 22 cases where $F_e$ is higher than 50 Wm$^{-2}$, which is a significant contribution to the boundary layer energy budget in shallow convective environments.

## 3.4 Modeled rain isotope ratios

### 3.4.1 P3 cases

We independently evaluate rain evaporation during ATOMIC by analyzing the change in precipitation isotope ratios ($\delta D_p$ and $\delta^{18}O_p$) from cloud base to the surface for the 22 P3 cases. Increase in $\delta D_p$ and $\delta^{18}O_p$ are modeled for drops 375-6500 $\mu$m diameters as shown in figure 11. Drops smaller or equal to 900 $\mu$m evaporate completely, and the bigger drops partially

evaporate before reaching the surface. The isotope ratios increase as the mass of raindrop decrease.

This is also shown by Salamalikis et al. (2016) who calculated the increase in $\delta D_p$ and $\delta^{18}O_p$ for drops 1 mm-3 mm in diameter falling from altitudes (≈1500 m) higher than this study (700 m). For a 1 mm raindrop, 70% surface relative humidity, and 303 K surface temperature, Salamalikis et al. suggest that the increase in $\delta D_p$ and $\delta^{18}O_p$ is 48‰ and 8‰, respectively (table 3, numerical isotope evaporation model in Salamalikis et al. 2016). Here, we find for a 1.1 mm raindrop $\delta D_p$ and $\delta^{18}O_p$

increase by 46‰ and 9‰, respectively. For bigger drops, the difference between Salamalikis et al. (2016) and this study is larger. For a 2 mm diameter drop, the net increase in $\delta D_p$ and $\delta^{18}O_p$ in Salamalikis et al. (2016) is 64‰ and 10‰, respectively, which is higher compared to 27‰ and 5‰, respectively, in this study. A case study in figure 2a of Salamalikis et al. (2016) where $\delta^{18}O_p$ is plotted with altitude for different raindrop sizes at temperature 278K and RH$_{sf}$ of 40% is replicated by our model to assess the consistency between our model and Salamalikis et al. (2016) (figure A6). The small inconsistencies might

be because the parameters used in our model are obtained from Graf (2017). Overall though, the two models are consistent especially for larger drops.

Because isotope ratios are typically measured in bulk precipitation, we evaluate the mass-weighted isotopic composition of the integrated raindrop size distribution in the simulations. This is done by integrating $\delta D_p$ and $\delta^{18}O_p$ over the observed RSD





for drops between 500 $\mu$m to 6 mm (17 bins) to estimate the mean $\delta D_p$ and $\delta^{18}O_p$ at each vertical level z following:

$$\delta_p(z) = \frac{\sum_{i=1}^{17} N_i D_i^3 \delta_{p,i}}{\sum_{i=1}^{17} N_i D_i^3} \tag{13}$$

When equation 12 is expanded in terms of a lognormal distribution (equation 11), $N_0$ cancels in the numerator and denominator making $\delta_p$ independent of raindrop concentration and only affected by $D_g$ and $\sigma$, much like the rain evaporated fraction (figure 7f). The RSD-integrated $\delta^{18}O_p$ and $\delta D_p$ are plotted for all 22 cases in supplementary figures 3-4, respectively. As shown there, the isotope ratios increase with rain evaporation in the sub-cloud layer. Eleven cases, including those on 4 February, show
complete rain evaporation within the sub-cloud layer since the drops observed are relatively small compared to other days (figure 3). The remaining 11 cases show increase in rain isotope ratios as they reach the surface.

The dependence of $\delta_p$ on $D_g$ and $\sigma$ but not on $N_0$ helps explain why an 'amount effect' (Dansgaard, 1964) is not always present in low-latitude isotopic data. The amount effect suggests that for a given rain sample, if the rain rate is high, then $\delta_p$ should be low, and vice versa. However, this may not always hold true for rain evaporation in all microphysical conditions. If
$N_0$ is large and $D_g$ and $\sigma$ are small, rain rate will be high, due to its strong sensitivity to $N_0$. But, $\delta_p$ will also be high, due to the small $D_g$ and $\sigma$. Similarly, if $N_0$ is small and $D_g$ and $\sigma$ are large, then both the rain rate and $\delta_p$ will be low. Consequently, the amount effect may not be appropriate for describing rain evaporation.

The $\delta D_p$ and $\delta^{18}O_p$ at cloud base prescribed from the in-situ isotope measurements for the 22 cases at cloud level vary within -12 to 9‰ and -2.5 to -0.45‰, respectively. The $\delta D_{va}$ and $\delta^{18}O_{va}$ for the 22 cases obtained from the nearest 150 m
level samples are -70±1‰ and -10.5±.05‰, respectively. In comparison, the modeled $\delta D_p$ and $\delta^{18}O_p$ at or near the surface vary within 5-35‰ and -1 to 7‰ respectively. For the four cases on 9 February, $\delta D_p$ and $\delta^{18}O_p$ increase roughly from -10 to 20‰ and -2 to 4‰, respectively, between cloud base and the surface (figure 7g-h).

The corresponding d-excess (= $\delta D_p$-8$\delta^{18}O_p$) decreases from 12‰ at cloud base to -14‰ at the surface. Because HDO evaporates more efficiently than $H_2^{18}O$ under non-equilibrium conditions, the d-excess of the raindrops decreases toward the
surface as evaporation occurs. The net decrease in d-excess between the cloud base and surface for the 11 cases where rain reaches the surface is also found to be proportional to the rain evaporated fraction at the surface (figure 7f and i). It is only weakly correlated with the reduction of rain rate or decrease in magnitude of $F_e$ (figure 7d,e and i). This suggests that d-excess can be useful in quantifying the fraction of rain evaporated in the sub-cloud layer.

### 3.4.2  Modeling surface observations (Ron Brown, BCO, Meteor)

The surface based observations of $\delta D_p$ and $\delta^{18}O_p$ from the Brown, the BCO and the Meteor are used to derive an observational estimate for d-excess in surface precipitation (figure 12). Meteor d-excess is slightly smaller compared to the BCO and the Brown, but overall the values are within 4-18‰ across the three sources (refer Bailey et al. (2022, submitted) for in-depth isotope data discussion). In general when isotope samples are collected, the surface rain rates measured on the Brown and at the BCO (1-5 mm/hr) are much more intense than those observed by the P3. The observed $RH_{sf}$ is also much higher: ($\approx$85%)
for the Brown compared to 70% for the P3. The higher rain rates and higher $RH_{sf}$ sampled by the Brown may be due to differences in the microphysical and thermodynamic properties of the rain sampled by P3 and the Brown.





We simulate the Brown d-excess observations (8-18‰) for various ranges of $D_g$ and $\sigma$ at 85% surface relative humidity (figure 13a) and use the rain evaporation model to diagnose the microphysical constraints required to replicate the Brown rain isotopic values. The ranges of $D_g$ (0.01-1 mm) and $\sigma$ (1-4) chosen for the d-excess simulation are quite broad and include values observed during the CSET and RICO campaigns. Note that d-excess is independent of $N_0$, so $N_0$ is not needed for the d-excess model simulation.

The model simulates the observed d-excess range from the Brown for multiple combinations of small $D_g$ and high $\sigma$ or high $D_g$ and small $\sigma$, as seen from the contour lines in figure 13a. For example, the 8‰ d-excess contour line is obtained for a combination of $D_g$ increasing from 0.01 to 1 mm and $\sigma$ decreasing from 3 to 1.54. While $D_g$ being higher than 0.8 mm is unlikely (the maximum CSET $D_g$ was 0.5 mm and the mean volume diameter at cloud base during a case study in RICO was 0.75 mm (Geoffroy et al., 2014)), the higher $\sigma$ values of 1.5-3 are plausible, as they are similar to values observed during ATOMIC and during CSET and RICO as well. The higher $D_g$ and $\sigma$ required to simulate the observed Brown d-excess at high $RH_{sf}$ also corresponds well with the higher surface rain rates observed by the Brown. The reproduction of the observed d-excess by the model verifies the appropriateness of the evaporation model for investigating precipitation process in the trade wind environment.

## 3.5 Comparing P3 and the Brown rain properties

Considering that $RH_{sf}$ for the P3 cases was much smaller (70%) compared to what was measured by the Brown (85%), we model the surface d-excess for $RH_{sf}$ of 70% for the same $D_g$ and $\sigma$ ranges as before (figure 13b). The ranges of $D_g$ and $\sigma$ observed during the 22 cases from the P3 are indicated by the red box in figure 13b. The corresponding d-excess values are mostly negative and much smaller than the d-excess observed at the surface by the Brown. This suggests that at lower $RH_{sf}$ or drier conditions, the amount of rain evaporation is much higher, even if the $D_g$ and $\sigma$ are large. For the same RSDs, the drier sub-cloud layer observed by the P3 would produce 20‰ smaller d-excess values than those observed by the Brown. This confirms the role of $RH_{sf}$ in controlling the rain evaporation processes.

Conversely, if the $RH_{sf}$ sampled by the Brown had been as dry as sampled by the P3 (70%), then the $D_g$ and $\sigma$ needed to simulate the observed Brown d-excess (> 8‰) would have had to have been improbably higher than observed by the P3. Furthermore, if $D_g$ for the Brown rain events were the same as for the P3 (0.2 mm), then $\sigma$ would have been higher than 2.5 to simulate the observed (>8‰) d-excess at the surface. This $\sigma$ is higher than the average $\sigma$ observed by the P3. Therefore, the $D_g$ at cloud base for the Brown rain events was likely higher than the $D_g$ observed by thr P3 (0.2 mm). Our study helps clarify the influence of microphysics on rain evaporation in shallow convective environments.

## 3.6 Water vapor isotope ratio variations

Because rainwater is isotopically distinguishable from background water vapor, we also consider whether raindrop evaporation can be detected as an isotopic change in the background vapor. However, for the 22 cases studied here, the increase in water vapor isotope ratios due to rain evaporation below clouds for both $\delta^{18}O_v$ and $\delta D_v$ is extremely low (<0.005‰ and <0.04‰ respectively). These variations are much smaller than the precision of the real-time commercial water vapor isotopic analyzers





deployed during ATOMIC, even when the data are averaged across an entire 10-minute level flight leg. This low increase is mainly due to small mass evaporated ($q_e$) compared to the total vapor $q_v$ ($\approx 15 g m^{-3}$). As shown by equation 9, the weight of the $\delta_{va}q_{va}$ term always dominates the $\delta_e q_e$ signal.

The effect of rain evaporation on the sub-cloud $T_a$ and $q_{va}$ is also negligible for the 9 February 150 m level RWC observations, where $T_a$ or $q_{va}$ anomalies for raining and non-raining samples do not show any distinguishable differences. This
suggests that the sub-cloud evaporation of rain on 9 February did not significantly change the background moisture and temperature.

## 4   Conclusions

This study evaluates shallow rain evaporation characteristics in the north Atlantic Ocean using a one-dimensional sub-cloud rain evaporation model initialized by in-situ observations from the ATOMIC campaign. The observed in-cloud leg mean rain
rates vary within 0.01-3 mm/day for 20 out of 22 cases sampled by the P3, with rates higher than 20 mm/day for the remaining two cases. The modeled $F_e$ over these 20 P3 legs also varies within 1-100 $Wm^{-2}$ and is comparable to the 3-day mean 100 $Wm^{-2}$ latent heat flux at 200 m altitude that was measured remotely from aboard the Maria S. Merian, another EUREC[4]A research vessel that sampled to the south of the P3 study region (Stevens et al., 2021), implying that rain evaporation is energetically significant within the Atlantic trade-wind region during ATOMIC.

Variability in the observed rain rates over the 22 cases is mostly due to the differences in microphysical characteristics at cloud base. Indeed, even though a decrease in $RH_{sf}$ is associated with an increase in rain rate, the effect is much weaker than the effect of the microphysics. The magnitude of modeled rain rates in the sub-cloud layer is particularly dependent on $N_0$, given the linear dependence of rain rate on $N_0$, compared to the more complex lognormal relationship with $D_g$ and $\sigma$.

Similar to rain rates, the magnitudes of the modeled $F_e$ in the sub-cloud layer also depend on $N_0$, i.e. the higher $N_0$ is, the
larger $F_e$ is at a given level. However, the vertical distribution of $F_e$ is dependent on variations in $D_g$ and $\sigma$. A higher $D_g$ or $\sigma$ at cloud base means higher concentrations of larger drops that are more likely to reach the surface without fully evaporating. This makes the vertical distribution of $F_e$ bottom heavy. A bottom-heavy $F_e$ profile (9 out of 22 cases) may promote the formation of cold-air pools near the surface and facilitate the advection of surface moisture upwards, a necessary process for cumulus formation. Conversely, a top-heavy $F_e$ profile (13 out of 22 cases) may create stability between the warmer cloud and cooler
sub-cloud layers and inhibit mixing.

Rain evaporation is also evaluated in this study using d-excess, calculated from hydrogen and oxygen isotope ratios in water. The changes in the modeled d-excess between cloud base and the surface for the P3 cases are proportional to the total percentage of rain evaporated in the sub-cloud layer. Consequently, a large decrease in d-excess from cloud base to the surface indicates that the percentage of rain evaporated in the sub-cloud layer is high. Both d-excess and the percentage of rain
evaporated are nearly independent of $N_0$ and depend sensitively on variations in $D_g$ and $\sigma$. Overall, the model suggests 65% of the rain evaporates in the sub-cloud layer for the P3 cases.





The surface d-excess measurements sampled by the Brown (8-18‰) are also used to assess the rain evaporation model at 85% $RH_{sf}$, which is a much higher value of relative humidity compared to the $\approx 70\%$ $RH_{sf}$ observed by the P3. The d-excess values observed by the Brown are simulated accurately at this surface relative humidity if $D_g$ and $\sigma$ are higher than observed

by the P3. This suggests that the RSD for rain sampled by the Brown had higher $D_g$ and $\sigma$ compared to rain sampled by the P3. This is also confirmed by the higher observed surface rain rates on the Brown, which were 1-5 mm/hr, compared to the 0.01-3 mm/day of in-cloud rain rates measured by the P3.

The influence of sub-cloud layer thermodynamics on rain evaporation is similarly evaluated by simulating the d-excess observed by the Brown for an $RH_{sf}$ of 70% . This experiment shows that if the Brown rain had occurred in drier conditions

($RH_{sf}$=70%), then the $D_g$ and $\sigma$ at cloud base needed to produce the observed surface d-excess (>8‰) would have had to have been much higher than physically reasonable for the shallow rain regime. The higher $RH_{sf}$ sampled by the Brown likely helped the rain at cloud base evaporate more slowly and eventually reach the surface. Similarly, higher $D_g$ and $\sigma$ at cloud base may have also enabled more drops to reach the surface. Thus, both the microphysical properties and thermodynamic conditions of the cloud and sub-cloud layers likely played important roles in controlling sub-cloud rain evaporation during ATOMIC.

In comparison to ATOMIC, the RICO campaign, which studied rain events in the vicinity of the P3 sample domain, had stronger rain events with higher RWC (2 $gm^{-3}$), higher $N_0$ ($10^4$ $m^{-3}$), larger mean volume diameters (0.75 mm) at cloud base, and larger $\sigma$ comparable to P3 observations (estimated from Geoffroy et al. (2014)). The CSET campaign over the Pacific Ocean also measured higher rain rates and larger $N_0$ ($10^4$ $m^{-3}$) but similar $D_g$ and $\sigma$ compared to ATOMIC. The lower magnitude rain rates sampled by the P3 compared to RICO and CSET may be due to the lower $N_0$ observed by the P3. This might be

due to the P3 sampling strategy, which prioritized cloud and aerosol measurements and avoided intense rain penetrations. Despite smaller rain rates, some P3 cases, especially those on 9 February, undergo significant rain evaporation. The rain evaporation model used in this study, while working accurately, ignores the effects of turbulence or mesoscale variability for simplicity. In general, the model performs well in depicting the importance of microphysical and thermodynamic processes on rain evaporation, documented through both isotope ratios and RSD, and would be helpful in evaluating the representation of

rain evaporation in more sophisticated cloud models.

*Acknowledgements.* This material is based upon work supported by the National Center for Atmospheric Research, which is a major facility sponsored by the National Science Foundation under Cooperative Agreement No. 1852977. The NSF support for EUREC4A-Iso campaign is from NSF grant No. 1937780. PNB's contribution to this work was supported by the National Science Foundation under Grant AGS-1938108. We acknowledge the entire team of ATOMIC/EUREC4A campaign for collecting and processing the data from all the platforms

used in this study. We thank the two internal reviewers at NCAR for their thoughtful comments.

*Data availability.* The description of the campaign is cataloged at https://psl.noaa.gov/atomic. The doi for all the processed datasets are available at https://psl.noaa.gov/atomic/data/.



*Author contributions.* MS, and AB designed the study. MS performed the analysis and and wrote the paper. AB, PB, SPD, DN and EQM revised the paper, provided important feedbacks on the figures and text. EQM processed the Ron Brown isotope datasets. ML and PC

collected all of the P3 microphysical datasets. ML stitched the CIP and PIP microphysical datasets for all the flights.

*Competing interests.* There are no competing interests.



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





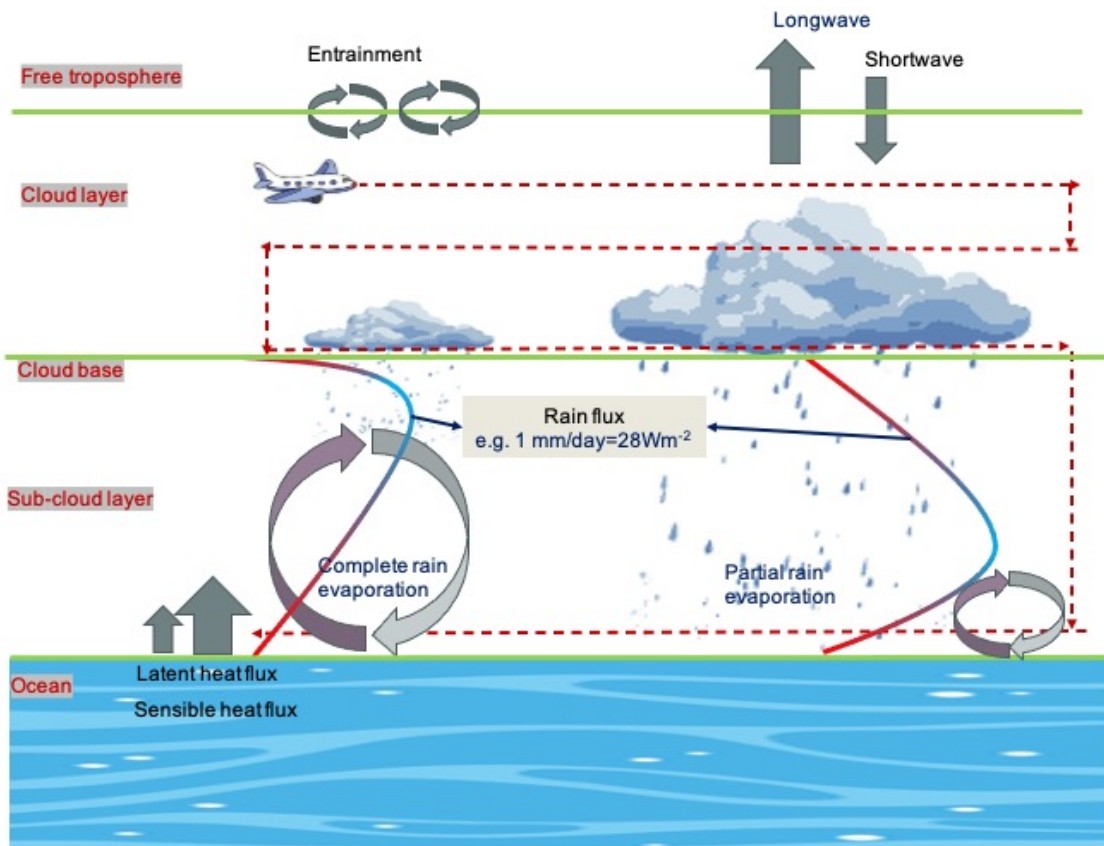

**Figure 1.** Schematic showing raining cumulus-topped boundary layer with surface latent and sensible heat fluxes, longwave and shortwave fluxes above cloud top, and entrainment mixing between free troposphere and cloud layer. Rain falling below cloud base can completely or partially evaporate, that can create different mixing configurations as shown by the arrow sizes and directions, and the shaded lines of vertical rain flux structures in the sub-cloud layer. Rain flux from 1 mm/day rain rate evaporation is shown as 28 $\text{Wm}^{-2}$. The aircraft measurements are made at horizontal above-cloud, in-cloud, cloud base and near surface legs as shown by air plane cartoon and red dashed-line trajectories.



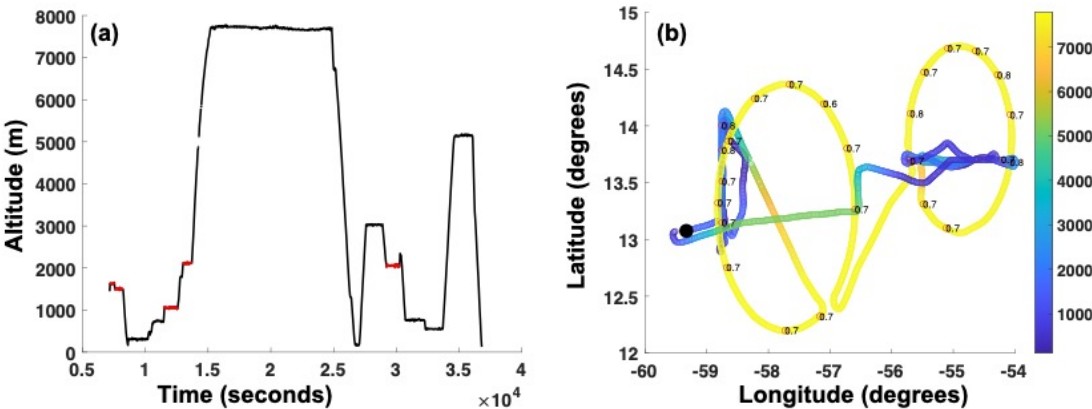

**Figure 2.** P3 trajectory on 9 February a) in time-altitude axes and b) in longitude-latitude axes with contour colors showing the altitude of P3. The red lines in a) denote the legs with mean rain rates greater than 0.01 mm/day that are selected for this study. Numbers in b) denote the surface relative humidity in fraction of 1 over the dropsonde locations.





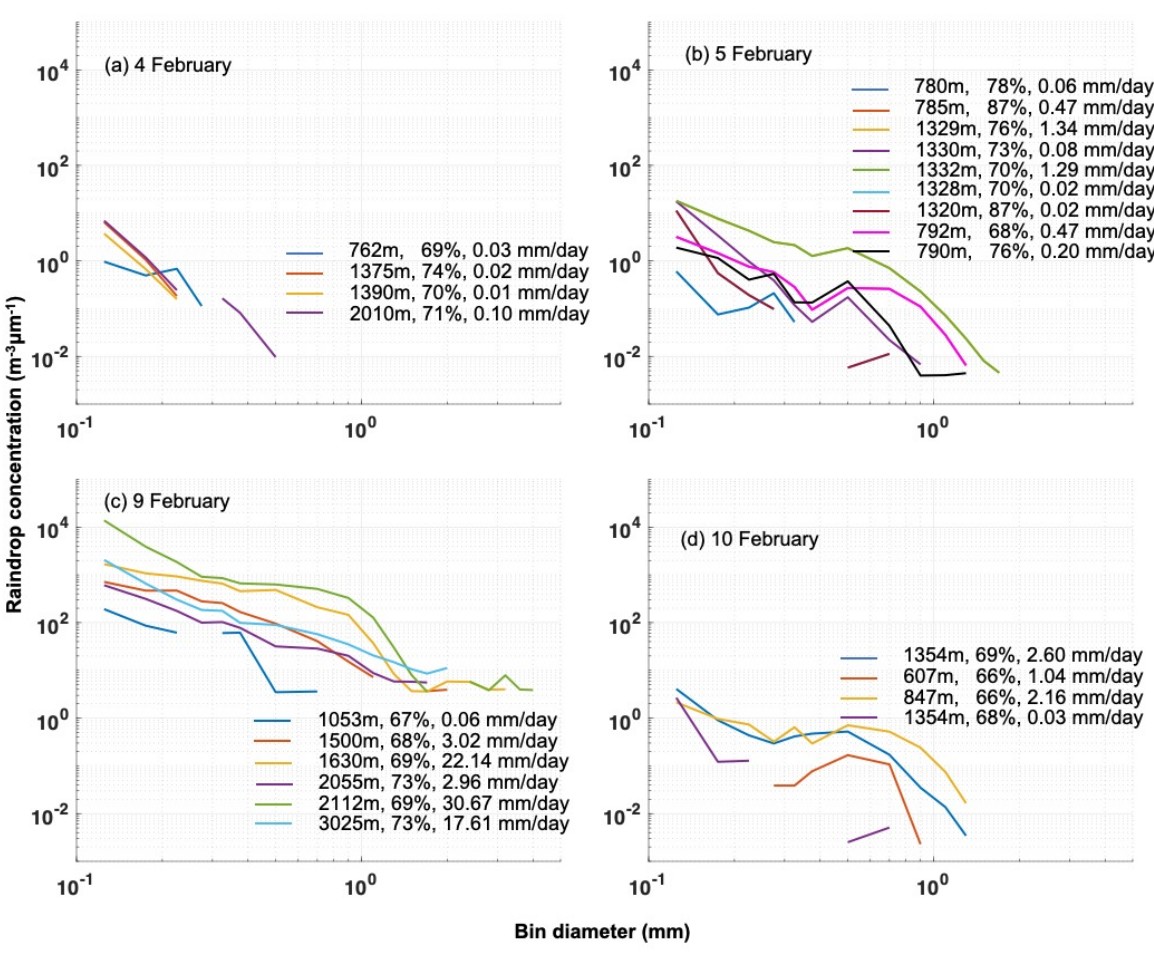

**Figure 3.** Raindrop size distribution is shown for 10-minute horizontal legs on a) 4 b) 5 c) 9 and d) 10 February. Legend shows the altitude of the horizontal legs, dropsonde-derived surface relative humidity and leg-mean rain rates.



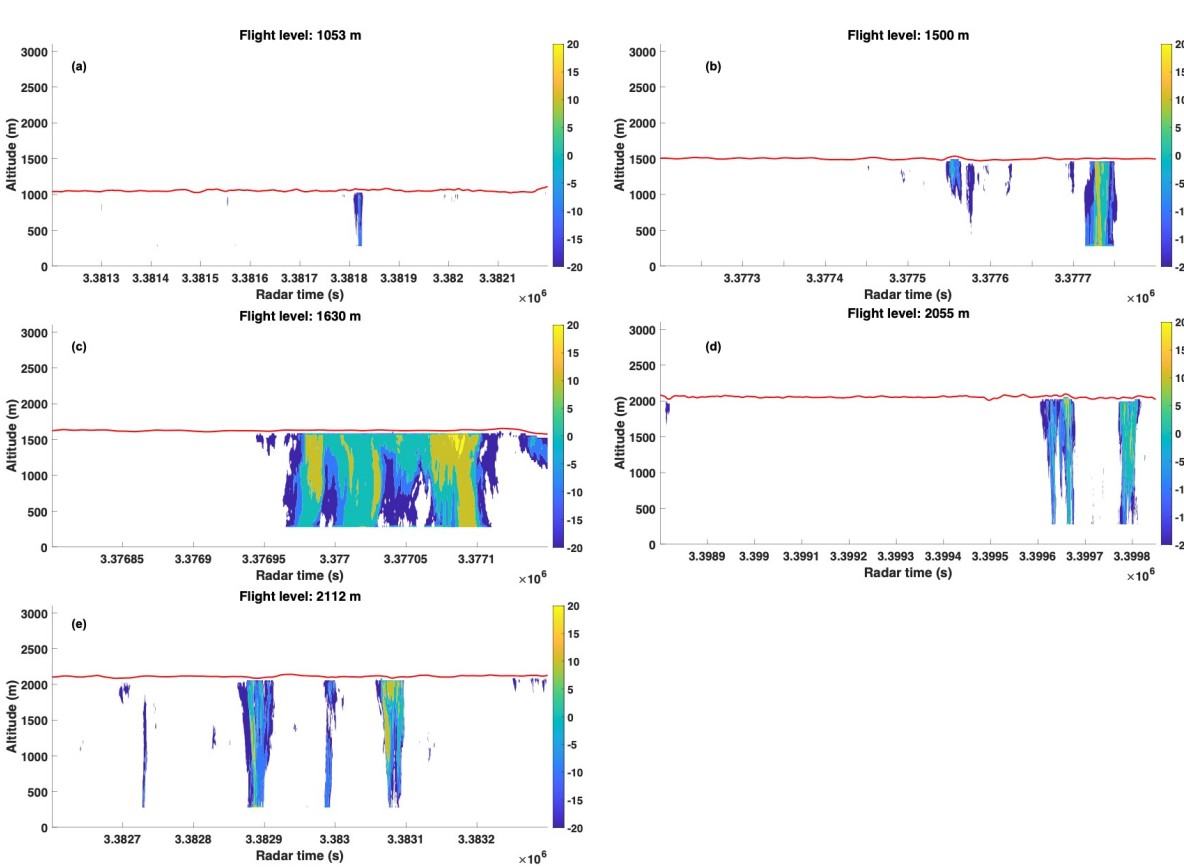

**Figure 4.** 94 GHz radar reflectivity in dBZ on board P3 on 9 February pointing downwards for all the six horizontal level legs shown in figure 2. Red line is the flight trajectory and contour colors show radar reflectivity.

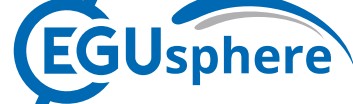

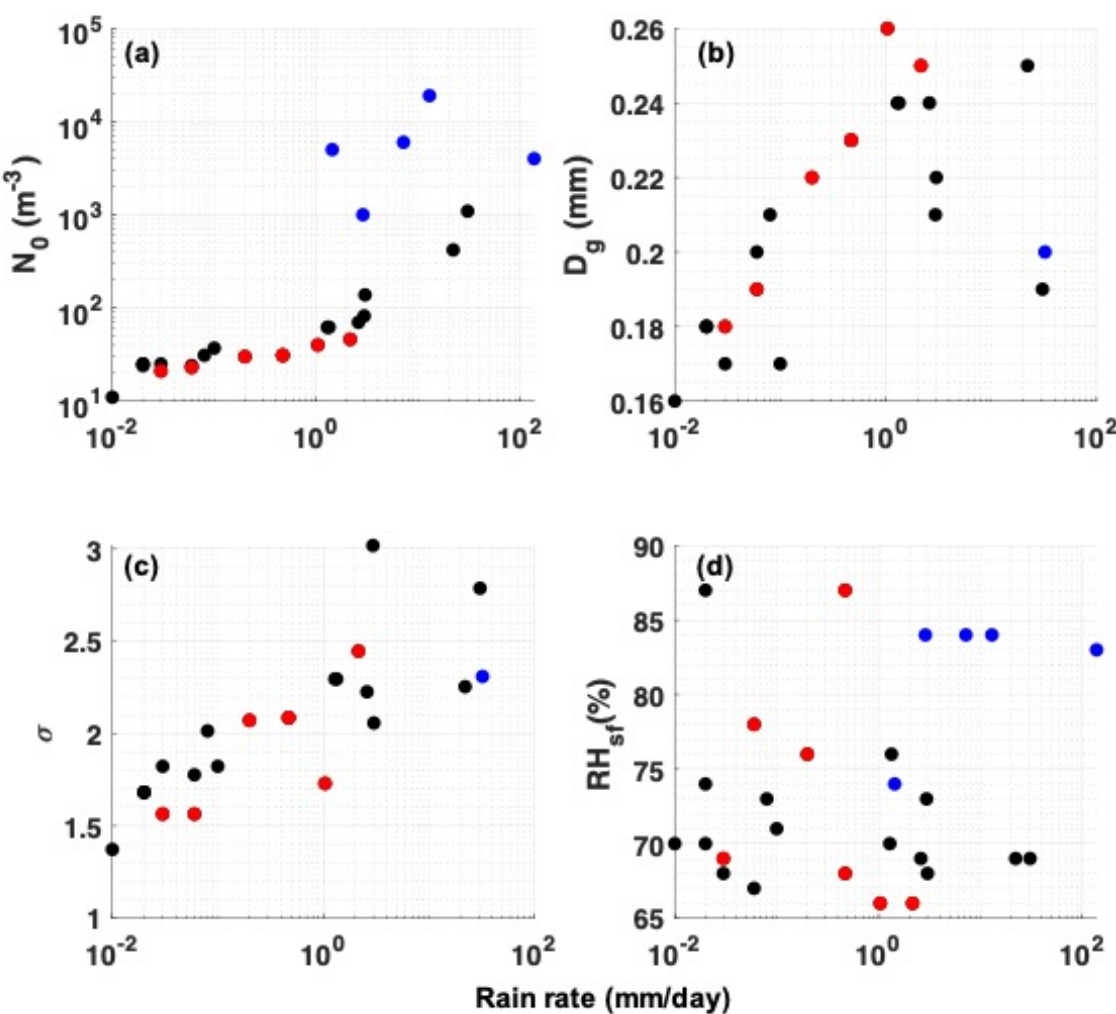

**Figure 5.** 10-minute leg mean rain rate is scattered against a) $N_0$, b) $D_g$, c) $\sigma$ and d) $RH_{sf}$ for 22 cases observed by P3. Red circles show the cases sampled at 700±100 m altitude, and black circles are cases sampled at higher altitudes. Blue circles are for CSET campaign obtained from Sarkar et al. (2020). Only the average $D_g$ and $\sigma$ over five CSET cases were available, and are shown by single dots in b-c). $N_0$ and $RH_{sf}$ were available for five cases during CSET as shown in a) and d).





**Figure 6.** Lognormally-fitted 1-Hz rain parameters (a,e,i,m) $N_0$, (b,f,j,n) $D_g$, (c,g,k,o) $\sigma$, and (d,h,l,p) rain rates are depicted as box-plots for all the 22 cases on (a-d) 4 Feb, (e-h) 5 Feb, (i-l) 9 Feb and (m-p) 10 Feb. The box plots denote 25th, 50th and 75th percentiles. The minimum and maximum extents of the whiskers denote the minimum and maximum data-points that are not an outlier. Outliers are shown in red '+' symbols. Outliers are considered to data points outside the +/-2.7×standard deviation and 99.3 percent coverage assuming that the data is normally distributed.



**Figure 7.** Modeled a)$\sigma$, b)$D_g$, c)$N_0$, d)Rain rate, e)$F_e$, f) fraction of rain evaporated, g) $\delta^{18}O_p$, h) $\delta D_p$, i) d-excess$_p$ vs. height for four cases (1500 m, 1630 m, 2055 m, 2112 m) on 9 February. The legend in e) show the RH$_{sf}$ and cumulative F$_e$ over the entire sub-cloud layer. Fraction of rain evaporated in f) is calculated using 1-$\frac{R_z}{R_{cb}}$ where R$_z$ and R$_{cb}$ are rain rates at an altitude z and at cloud base.



**Figure 8.** Vertical profiles of modeled $F_e$ for 50 m vertical resolution as a function of a) surface relative humidity, b) $N_0$, c) $D_g$ and d) $\sigma$. The sub-cloud cumulative evaporation flux ($F_e$) is mentioned in the legend. Note the difference in x-axis scales for a) vs. b-d).





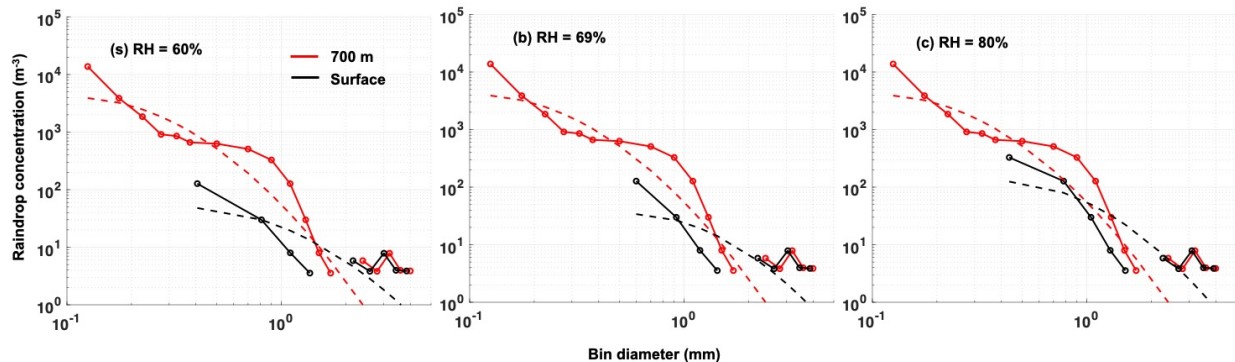

**Figure 9.** Raindrop size distribution at 2112 m case on 9 February with the highest rain rate are shown in solid red lines for RH$_{sf}$ of a)60% b)69% and c)80%. The modeled raindrop size distributions at surface are shown in solid black lines. The dashed red and black lines are the lognormally fitted lines at cloud base and surface respectively.



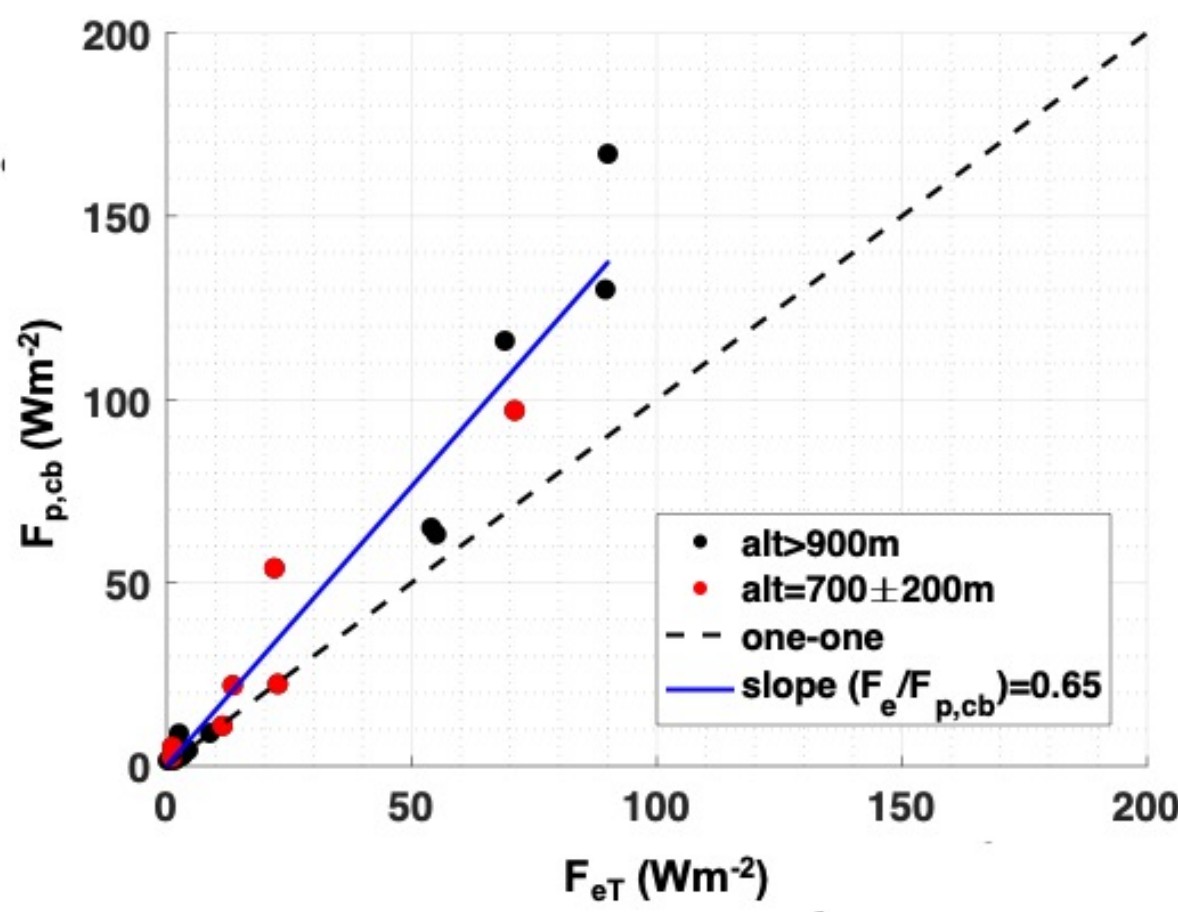

**Figure 10.** Rain flux ($F_p$) at cloud base is scattered against rain evaporation flux ($F_e$) for 20 cases sampled by P3. The two cases on 9 February (1630 m, 2112 m) where $F_e$ is higher than 500 Wm$^{-2}$ are not included. Red circles are cases sampled at 700±100 m altitude and black circles were sampled at higher altitudes. Blue line is the slope through the 20 cases and is 0.65. The dashed black line is one-on-one ratio line for reference.



**Figure 11.** Modeled raindrop diameter-resolved $\delta D_p$ and $\delta^{18}O_p$ plotted against altitude for vertical grid resolution of 1 m (top) and 50 m (bottom).



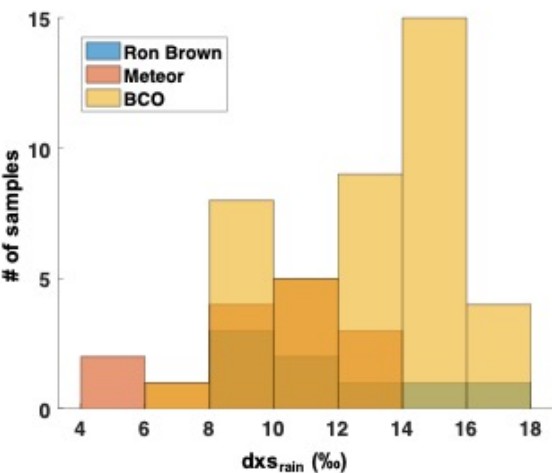

**Figure 12.** Histograms of observed d-excess at surface sampled by the Brown, Meteor and BCO.

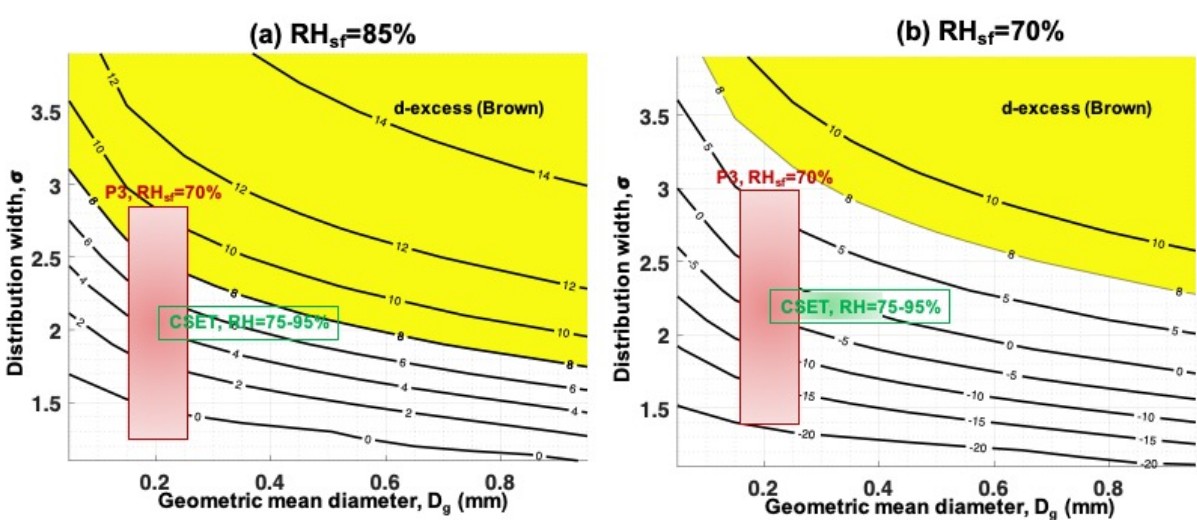

**Figure 13.** Contours of modeled rain d-excess at surface is shown as a function of $D_g$ and $\sigma$ for $RH_{sf}$ of a) 85% and b) 70%. d-excess higher than 8‰ were observed by the Brown and are shown in yellow shaded contours. The $D_g$, $\sigma$ and $RH_{sf}$ observed during 22 cases of P3 and during CSET campaign are shown in red and green boxes respectively in b).




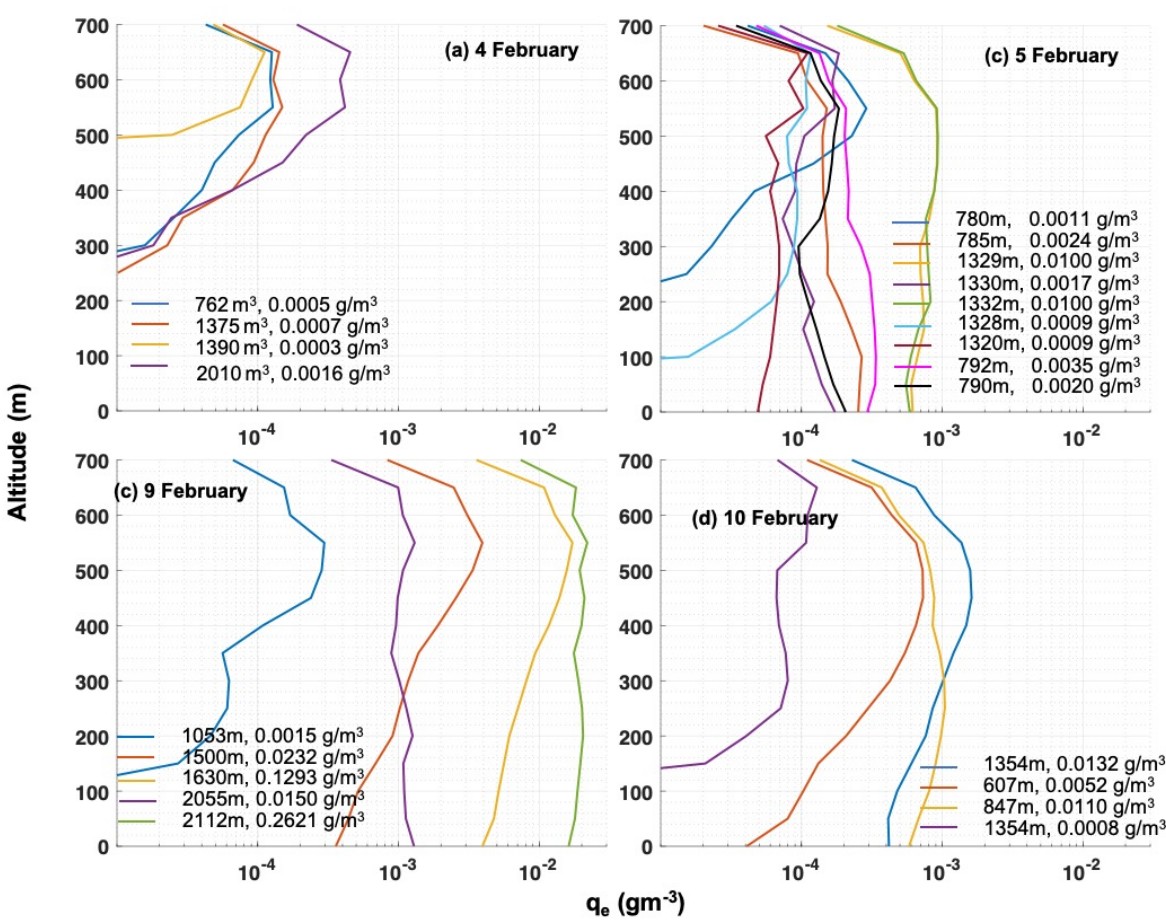

**Figure A1.** Modeled rain water content evaporated ($q_e$) between levels separated by 50 m vertical space plotted against altitude for 22 cases during a) 4, b) 5, c) 9 and d) 10 February. The legends show the altitude of each case followed by the difference of RWC at 700 m and surface ($q_{eT}$).



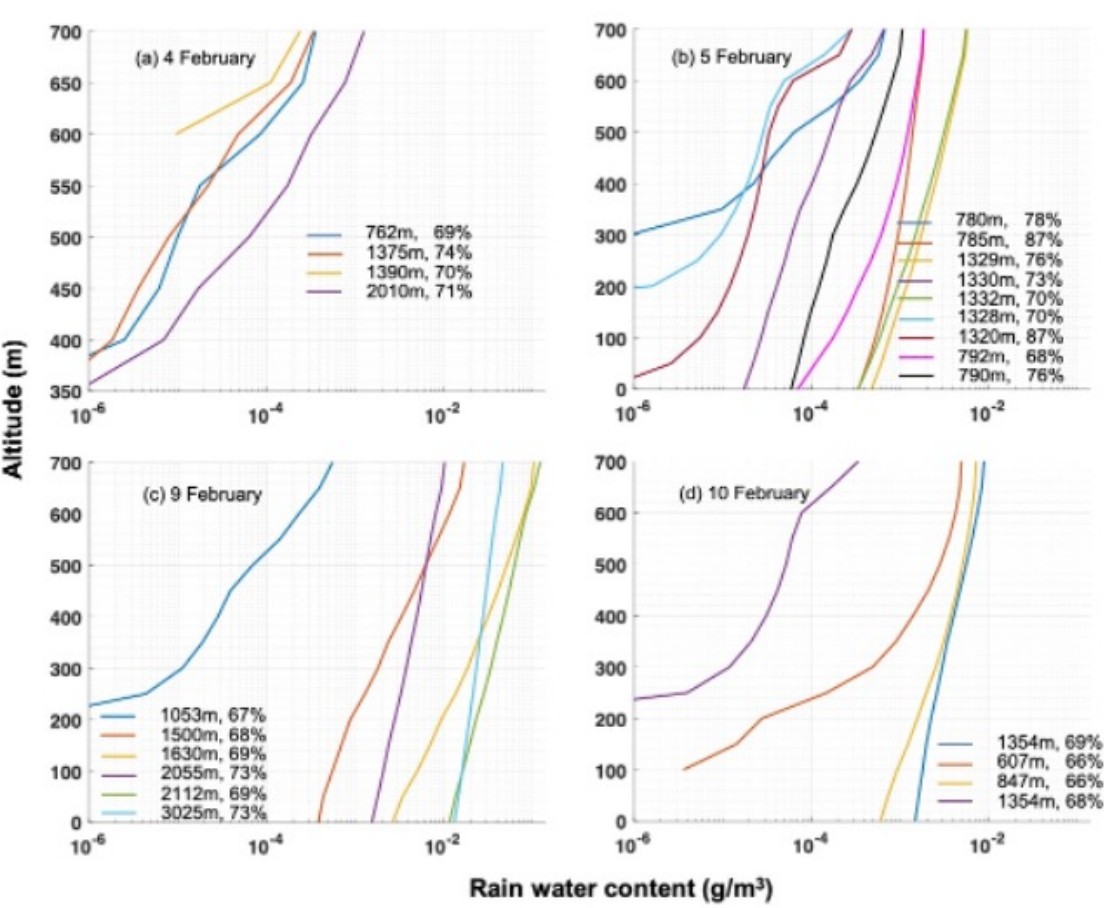

**Figure A2.** Modeled rain water content (RWC) plotted against altitude for 22 cases during a) 4, b) 5, c) 9 and d) 10 February. The legends show the altitude of each case followed by the surface relative humidity.





**Figure A3.** Modeled rain evaporation flux ($F_e$) plotted against altitude for 22 cases during a) 4, b) 5, c) 9 and d) 10 February. The legends show the altitude of each case followed by the difference of rain flux ($F_p$) at 700 m and surface ($F_{eT}$).





**Figure A4.** Modeled $\delta D_p$ plotted against altitude for 22 cases during a) 4, b) 5, c) 9 and d) 10 February. The legends show the altitude of each case followed by the difference in $\delta D_p$ between cloud base and surface.







**Figure A5.** Modeled $\delta^{18}O_p$ plotted against altitude for 22 cases during a) 4, b) 5, c) 9 and d) 10 February. The legends show the altitude of each case followed by the difference in $\delta^{18}O_p$ between cloud base and surface.



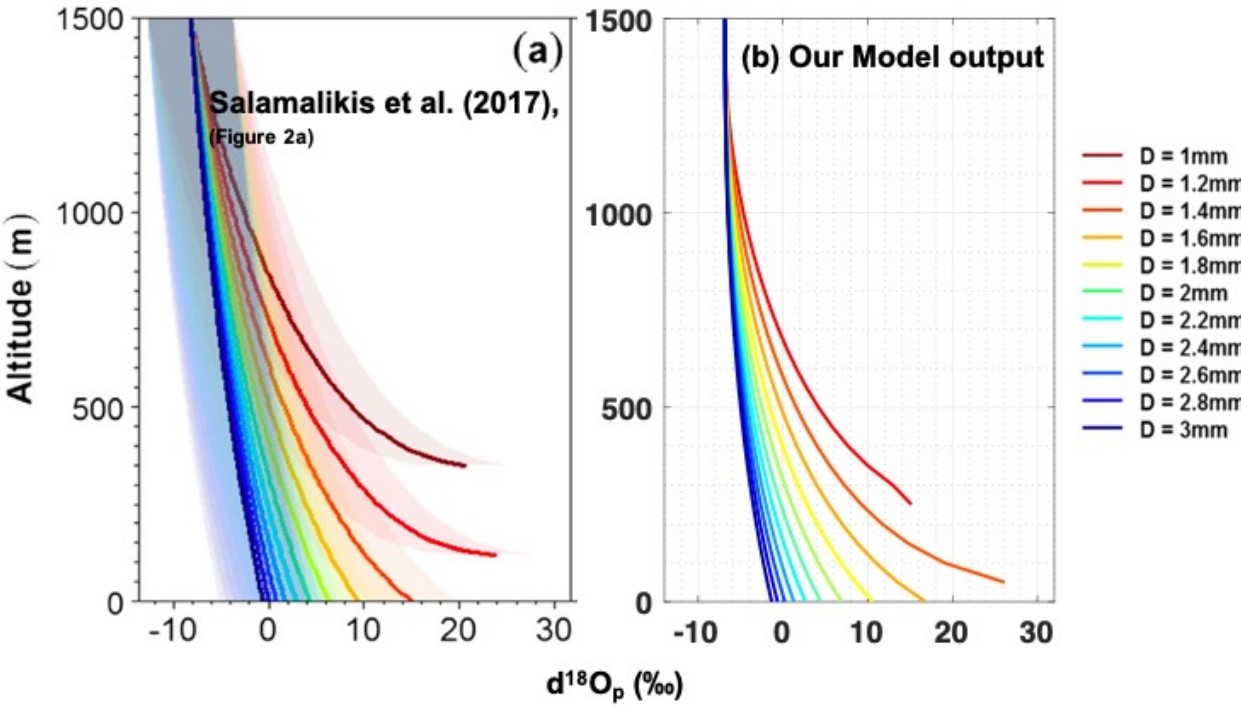

**Figure A6.** Modeled $\delta^{18}O_p$ plotted against altitude for $T_a$=278K and $RH_{surf}$=40% (a) as shown in Salamalikis et al. (2016), figure 2a (b) as obtained from our model.