# Peer review of "Sub-cloud rain evaporation in the North Atlantic winter trades derived by pairing isotopic data with a bin-resolved microphysical model"

_EGUsphere, 2022_

## Referee Comment (RC2)

This paper uses observations from the 2020 EUREC[4]A field campaign in combination with a one-dimensional model that simulates the change in drop size and isotope composition of the drops for given initial conditions that are constrained by aircraft observations (raindrop size distribution, isotope composition of the drops). I very much enjoyed reading this nicely written paper. Below cloud evaporation is a strongly under-researched topic. Since it is one of the two components of precipitation efficiency (together with conversion efficiency in clouds), the lack of constraints on below cloud evaporation from observations has important consequences for our ability to correctly predict precipitation at the weather event as well as the climate timescale.

I have four major comments, listed below, as well as a short list of minor/technical comments below:

Major comments:
A) It would be very valuable to point out the more general implications of this work already in the introduction (as well as at the end of the conclusion) e.g. trying to constrain precipitation efficiency with isotope observations would be one aspect to add to the now rather narrowly focused introduction. Of course, there are other aspects such as the impact of moisture recycling on mesoscale organization, which might be what the authors are more interested in. Right now the introduction reads like a nice summary of reference values of subcloud layer rain evaporation $F_e$ in terms of energy input into the subcloud layer, but these values would be much more interesting to compare with the author's results later on in the discussion.
B) Given the motivation of the authors to investigate below cloud evaporation, because it represents a substantial energy and humidity input into the subcloud layer, I think that the results from Section 3.6 are very disappointing. Can this aspect be discussed in more details? It seems unlikely that such an important process would leave no distinguishable isotope signal in ambient vapour. Do these results imply that, even though below cloud evaporation strongly impacts the amount of precipitation that reaches the surface, for the subcloud layer moisture budget, it is a negligible process? Or is it only important, when integrated over larger spatio-temporal scales than a single precipitating cell? How much does the authors' finding depend on the uncertainties of the aircraft and ship-based observations? Also can this aspect really be assessed with the model at hand given the assumption that vapour contributed by rain evaporation is neglected (L. 125).
C) There is very limited literature about below cloud evaporation effects, I agree, but I think there are a few studies from different settings with which the results in this paper can be compared to. For example, Aemisegger et al. 2015 GRL used a combination of numerical experiments and isotope observations to assess the importance of below cloud evaporation for a cold front passage. They found that over the whole frontal precipitation event neglecting below cloud evaporation leads to depletion biases of 20–40‰ in d2Hp and 5–10‰ in d18Op as well as to an increase of 74% in rainfall amount. This impact on total rainfall amount is very close to what the authors find in their study over the tropical North Atlantic. Also, in this paper, a substantial impact of below cloud evaporation on ambient vapour was found. How comes that in the winter trades the impact is so small? (Different region, different dynamics).

D) The discussion of the impact of $F_e$ on stability is interesting but confusing. The statements at L. 274ff and in the conclusions (L. 432) are contradicting. Please clarify. I don't understand, based on which of their findings they draw these conclusions on stability. If $F_e$ has limited to no impact on subcloud layer temperature and specific humidity (see also major comment B) then how can stability be impacted?

Technical comments:
1) I think ACP titles are usually not capitalized and I would strongly encourage the authors to mention more specifically the type of precipitation events they are looking at: "Sub-cloud rain evaporation from shallow convection in the North Atlantic winter trades"
2) The variables should be in italics except for abbreviations such as RH.
3) L. 5 not sure I immediately understand, in which phase dD and d18O were measured and used in the model
4) L. 9: 65% of what? mass, volume, event duration, number of events?
5) L. 17: is precipitation in shallow convection regimes really "ubiquitous"? I would have said it is rather sparse with low precipitation efficiencies compared to other cloud systems?
6) L. 48: "facilitate or hinder boundary layer stability" sounds a bit strange, how about "Does Fe reinforce or weaken the subcloud layer stability?"
7) L. 58-60: "This is because as rain evaporates…". I am not sure I can follow the implication that is formulated in this sentence. Vapour isotopes can be used independently to assess rain evaporation because rain evaporation leads to an enrichment of rain? This also joins my major point B above.
8) L. 114: degree W formatting
9) L. 124: what is the implication of ignoring collision-coalescence for your results?
10) L. 125: This last sentence leads to confusion about how you can assess the impact of below cloud rain-vapour interaction with the chosen 1D modelling approach.
11) L. 134: could the variables be described one after the author, instead of the long list of variables and then a long list of descriptions? (would be easier for a reader like me to grasp).
12) L. 150: "." Formatting, should go to L. 149.
13) L. 161: all parameters from Graf 2017: this is a bit vague. Which exactly and is this a good choice given the large contrasts between a cold front in the midlatitudes and shallow convection in the tropics? Maybe a summarizing table in the Appendix would help.
14) P. 5: what is the impact of RWC estimates from P3 observations on the modelling? How do they compare to the ATR observations from which large statistics are available at the cloud base level. See Bony et al. 2022 ESSD.
15) L. 184: this seems to contradict the statement at L. 125. Wouldn't an integration over longer time intervals (precipitation events of 10-30 min) be necessary to assess this aspect?
16) L. 274ff: is this speculative, or based on some specific results? And what dominates for changing stability: the evaporative cooling or the moistening effect (which are counteracting each other)?
17) L. 285-294: I like this finding a lot. Very clearly explained!

18) L. 357: Also this finding is very interesting! Maybe: "Consequently, the amount effect may not be appropriate for describing the impact of rain evaporation on the isotope composition of rain"? Did the authors' consider using the Graf et al. 2019 ACP delta-deltaD vs. delta-dexcess phase space for assessing the impact of different N0, cloud base RWC, RHsf etc.?

19) L. 415: this suggests as a consequence that the impact of sub-cloud evaporation on stability is negligible too (see again my major point B).

20) L. 418: **N**orth Atlantic

21) Conclusions: could the authors point towards bigger implications and open up on further research that could be done e.g. for better constraining below cloud evaporation in models?

---

## Author Comment (AC1)

**Response to referee comments on "Sub-cloud rain evaporation from shallow convection in the north Atlantic winter trades" by M. Sarkar, A. Bailey, P. Blossey, S. d. Szoeke, D. Noone, E. Q. Melendez, M. Leandro and P. Chuang.**

We thank both the referees for their recommendations and comments on the manuscript. We believe these feedbacks have made our manuscript more comprehensive and informative. The referee comments are shown in blue and author answers in black.

Reply to Referee #1 comments:

Major changes have been incorporated throughout the manuscript since the last submission in terms of paragraph organization and story-telling. The introduction and conclusions are rewritten to make the paragraphs tighter and clearer. The broader implications of this work are also discussed in the introduction. The comparison with previous campaigns has been removed from the introduction and discussed more elaborately in the results and conclusions sections. Some figures have been removed, others improved, to better represent the theme of our work.

In addition to some unit corrections in the code, the vertical resolution is also improved. Instead of a fixed 50 m vertical resolution, a nominal step size of 1 m is used. In addition, an adaptive step size is employed to make the model stable and adaptive for very small droplets. The MATLAB code function is made available to the ACP for publication. The improvements have led to an increase in the overall rate of evaporation. This has increased the rain evaporation fluxes and the surface d-excess of rain.

**General comments:**

1. The model formulation by Graf et al. 2017 has not been tested using aircraft observations of microphysics and atmospheric thermodynamic profiles to my knowledge. Previously observations of vapor and precip in these types of analysis were limited to near-surface observations. Often there are many unconstrained parameters such as the droplet sizes, lower tropospheric relative humidity, and rain and vapor isotopic compositions. The unique initialization and sensitivity tests that the authors do are valuable. However, it is difficult to discern the contribution of this effort to the field in its current form.

Our work provides observationally estimated analysis of sub-cloud rain evaporation, which we believe, would improve our overall knowledge of the shallow cloud processes.

The model formulations such as used in Graf et al. 2017 have indeed not been tested using aircraft microphysical and thermodynamic observations. This makes our work distinct and novel since we incorporate the high-resolution microphysical rain data from the aircraft to study rain evaporation. Following this, we also present a novel way to estimate the fraction of rain evaporated, rain evaporated flux in the shallow rain regimes, and their dependence on the microphysical and thermodynamic factors using aircraft observations, something that has not been done before to our knowledge. These estimates provide us a base to compare them with other rain regimes and from climate models.

Further, the rain evaporation budgets have been compared with those roughly estimated from other field campaigns. This brings perspective to the importance of sub-cloud rain evaporation in shallow clouds compared to other components of BL fluxes.

This study also highlights the importance of microphysical processes as opposed to the thermodynamic processes alone. The role of thermodynamics in effecting rain evaporation has been well established by previous works, such as Worden et al. 2007 and Risi et al. 2021. This could be done due to the ease of availability of thermodynamic observations in the sub-cloud layer.
However, our study clearly indicates how the thermodynamic processes alone would not be sufficient in explaining the rain evaporation processes. The results indicated how the microphysical processes in the sub-cloud layer are also influential in effecting rain evaporation. This was solely possible due to the rain evaporation model constrained by the microphysical datasets from the aircraft.

2. There is a lot of hand waving about whether the model constrained by the P3 observations can reproduce rain isotope values at the surface samples from the Ron Brown. The authors show that the model does not reproduce d-excess at the surface without a large change in relative humidity assumptions or larger drop sizes at cloud base. What do we learn about the representativeness of the model from this analysis?

The improvement in the model now shows that the modeled isotope ratios closely match the surface observations so long as the variations in relative humidity conditions are accounted for. We increase the $RH_{sf}$ of all the P3 cases to 85% to match the $RH_{sf}$ from the Brown observations. Doing so increases the modeled d-excess for the P3 to match those from the Brown.

Further, two cases from the P3 for which observed $RH_{sf}$ are 86%, have the modeled d-excess of 9 permil which matches the surface-based d-excess observations very well. All this lends credibility to our model and emphasizes that if the initialized $RH_{sf}$ in the model is accurate, then the model outputs would be accurate as well. Therefore, the rain evaporation analysis can sufficiently rely on the model provided that the $RH_{sf}$ is defined well. This has been explained further in the manuscript with the help of the new model results.

3. Fig 13 is an important 'take-away' figure, but it is difficult to understand. The model is challenged to reproduce surface rain d-excess values of >8 permil. I'm trying to find what the starting cloud-base d-excess values are based on the P3 observations. Fig 7 is the only thing I can find and that shows model values of ~10 permil. Little change in rain drop d-excess would suggest very small rain evaporation rates. Is it possible that d-excess is not a strong indicator of evap rates? Would dD or d18O be more sensitive? Many of the figures show vertical profiles of dD or d18O, but then the model is only tested against d-excess at the surface. How does it perform against d18O and dD?

The d-excess is more sensitive to the rain evaporation compared to d18O or dD alone. This is because while the variations in dD and d18O could be due to both equilibrium and kinetic fractionations, the d-excess cancels out the covariations in dD and d18O due to equilibrium fractionations. Rain evaporation, which is essentially a non-equilibrium process, is therefore more suited to be analyzed using d-excess.

This becomes clearer from the histograms in figure 10 for dD, d18O and d-excess. The effect of rain evaporation (through changes in $RH_{sf}$) is evident in d-excess. At low RH, d-excess is small and vice-versa. Comparatively, the effect of RH is less distinct, especially for dD, for which the histograms overlap. This makes d-excess more to study rain evaporation compared to dD or d18O alone.

The isotope ratio information used to initialize the model at cloud base has now been described in the caption of figure 10 (earlier figure 13). The vertical profile of dD, d18O and d-excess for all the 22 cases are included in figure 7g-i. We hope this makes our isotope analysis clearer.

What about the percentage of rain evaporated? The abstract sets up a relationship between the percent rain evaporated and d-excess, but this figure doesn't demonstrate that link or how different the percent evaporated estimates may be in the different cases.

The relationship between the percentage of rain evaporated and the d-excess is now shown in the appendix figure A1. The $F_{eT}/F_p$ or REF is the fraction of rain evaporated, and it is proportional to the fractional change in d-excess defined as $1-(d_{p,sf}/d_{p,cb})$ over the P3 cases where rain reach the surface.

The fit has RMSE=0.18 and SSE=0.6 with a polynomial equation of:

REF = $p1*(1-(d_{p,sf}/d_{p,cb}))+ p2$

Coefficients (with 95% confidence bounds):

   p1 =    0.5941  (0.2289, 0.9594)

   p2 =     0.182  (-0.1298, 0.4937)

In this way, the fraction of rain evaporated in the sub-cloud layer could be estimated from the change in d-excess.

   4.  One important observation in the abstract concerns the vertical structure of rain evaporation which is sensitive to the droplet size distributions rather than the droplet concentrations. Is this droplet-resolving model unique in that regard? In other words, does the non-isotopic information provide any valuable insight as well?

We are not aware of any previous study that has looked into the sensitivity of the vertical rain evaporation structure to the microphysical parameters. However, LES studies in Sandu et al. 2011 are conducted that show how increased precipitation at cloud base could affect the mixing state of the sub-cloud layer. The conceptual models in Paluch & Lenschow 1991 and those discussed in Srivastava 1985, also describe how the rain evaporation could affect the vertical temperature profile in the marine boundary layer. However, these works only look at how the thermodynamic changes affect the BL.

Comparatively, in our work, we were able to delve into both the microphysical and thermodynamic effects on the rain evaporation. This was made possible by the drop-resolved feature of the model. The model results show how strongly the vertical structure of the rain evaporation is linked to Dg, sigma (shape), and not to N0 (magnitude) of the raindrop size distribution (RSD) in figure 8b-d. Together, the model provides insight on the importance of cloud base microphysical properties to the rain evaporation processes in the sub-cloud layer.

   5.  How do the isotope observations improve understanding compared to other methods used in the field?

One important perspective that the isotope observations give us is that the change in d-excess across the sub-cloud layer is indicative of the fraction of rain evaporated. This correlation could be utilized by comparing the aircraft isotope measurements at cloud base and surface to obtain a rain evaporation estimate. This isotope-inferred estimate could be compared with microphysically-inferred estimate to obtain two independent estimates of rain evaporation. An independent isotope estimated rain evaporation is also useful because microphysical measurements are pretty uncertain. The isotope perspective would help in reducing observational errors and making the isotope-microphysical derived relationships more robust.

   6.  Title could be improved by mentioning the model and observations.

The title has now been changed to "Sub-cloud rain evaporation from shallow convection in the north Atlantic winter trades".

7. Overall, there are many figures. Are they all important for telling your story?

Some of the figures in the original manuscript has been removed, reducing the total number of figures from 13 to 10. Among these are some new figures and some improvements on the old figures which we believe represent the manuscript more concisely. Three figures are also being added in the appendix for additional details.

8. One valuable contribution that the authors could provide is making this vertically-resolved model publicly available. I encourage the authors to share their code with the community. It could be useful for providing Monte-Carlo estimates of surface rain isotopic composition in future studies.

We completely agree on this. The code will now be made available along with the manuscript.

**Specific comments:**

Consistent unit notation needed throughout. E.g. mm day-1, W m-2

Done.

Line 20: Which is more common in the field to describe? Evaporative flux or latent heat flux?

Generally latent heat flux is used to describe the flux due to any phase change. But since phase change can be from condensation as well as evaporation, we have clarified the sentence further as "Rain rates on the scale of 1 mm/day, commonly associated with shallow cumulus precipitation, are capable of producing roughly 28 Wm-2 of latent heat flux through rain evaporation in the sub-cloud layer…". L21

Line 60: 'its' can be unclear. Edit to mention rain drop isotopic enrichment

The sentence has been further simplified as "This is because as rain evaporates into the unsaturated sub-cloud layer, the isotopically light water transitions to the vapor phase more efficiently, causing the drops to become increasingly heavy (Salamalikis et al., 2016; Graf et al., 2019)." L58

Line 61: define RSD at first use

The RSD is defined at its first use in the new version of the manuscript.

Line 87: is there a citation for the reliable/unreliable size ranges?

There are no specific citations available for the ranges. information on the reliability of the drop size ranges is obtained through direct correspondence with the data authors (Leandro and Chuang, 2021) who are also the co-authors of this paper. The details on the size ranges of the CIP and PIP instruments could be referred from the website: https://www.dropletmeasurement.com/.

Line 92: provide units for parameters in the equation.

Done. The new sentence reads "The rain rates (in mm/day) are calculated from the observed RSD using R=….."L98

Line 106-107: consider moving this sentence before "During ATOMIC"

The sentence has now been moved to the beginning.  L112

Line 117: Mention or cite Picarro calibration and data correction.

The Picarro uncertainties are now mentioned. L109 and 114.

Line 125: Several assumptions are made here. What implications does this have? In what way is the system in steady state? Equations 9 and 10 contain terms for the vapor from rain evaporation. Why state that it's neglected?

The line stating that the vapor contribution is neglected was confusing and is now removed. The steady state condition of the model implies that no new vapor source is introduced in the model, and that the ambient vapor already includes the evaporated vapor from rain that has taken place already.

The other assumption about ignoring any collision-coalescence process or drop interaction between the aircraft sampling altitude and the cloud base is now defined in L229. The assumption is backed by the similarity in the microphysical parameters that is seen for samples closer to cloud base and those higher up in figure 5.

Eqns 1 and 2: Were dD/dz and dTr/dz calculated for each diameter bin?

Yes, the dD/dz and dTr/dz are computed for each diameter bin. This is now explained from L142-168.

Line 129-130: Can you include some of the dropsonde data that confirms linear decrease in RH through the atm in a SI figure?

A new figure A3 shows the linear decrease of RH with height for all the dropsondes from the P3.

Line 133: Cite source of Eqn 1?

Done.

Line 134: list parameters and names one at a time so it's easier to match up.

Done.

Eqn 3: might help if it's shown as RWC(z).

The RWC term in equation 3 is now described as being calculated at vertical level z (L169).

Equations: For all equations that are evaluated at altitude (z) steps or bin sizes (i), write the equations indicating that.

All the equations have been modified accordingly.

Line 149: is L defined somewhere?

L is defined at L164.

Line 160: was the assumption that the BL was well mixed and 150 m delta_vapor observations are representative of the BL supported by the other observations?

The BL mixing is supported by the dropsonde profiles where the average specific humidity decreases from 15 g/kg to 13 g/kg from surface to 700 m (refer to Figure A3).

Line 162: All parameters obtained from Graf 2017 except the drop sizes.

The improved code includes parameters from Graf 2017, Salamalikis et al. 2016 as well as Pruppacher and Klett-2010.

Line 167: "validate the accuracy of the model" might be a reach given the current conclusions.

The improved model brings the modeled values closer to the observations and therefore increasing the accuracy of the model. This has been discussed further in section 3.4.2.

Line 170-174: I'm having a hard time understanding how Eqn 9 is calculated. qe is considered negligible, so qv = qva? qva is assumed constant but qv is calculated every 50 m? I'm getting stuck on what is allowed to change in the model, but doesn't change much verses what isn't allowed to change in the model.

In the revised manuscript, we have clarified how qe is not negligible, and how to determine the evaporated rain concentration by using Fe and moisture accumulation time. This part is in sections 2.3 and 3.5.

Additionally, delta_va is kept constant for the delta_p calculation (equation 10). This is done under the steady state assumption of the model where rain evaporation is assumed to already have taken place, and so the ambient vapor delta_va includes the effects of the rain evaporation already.

Eqn 10: Earlier it was mentioned that delta_v doesn't change with altitude?

δva does not change with altitude, but δv should.

Methods: This system of equations has parameters that feed back onto other equations. How were these solved at steady state? Iteratively? Please provide details.

Done. Line 142-168.

Line 198: define rain frequency metric

Done. L239.

Line 225: location of the RICO campaign?

The location of the RICO campaign along with CSET and ASTEX is now given in the introduction line 37.

Line 234 and Fig 5d: I do not see the negative correlation between RH and rain rate. Can you provide statistical evidence? This seems contrary to expectations.

The negative correlation is based only on the P3 datasets (red and black circles in figure 5d). The slope is negative with SSE=0.06. We speculate that the negative correlation could be due to the drier airmass from the free troposphere that could be reaching the surface through the downdrafts making the surface drier and reducing RH at surface.

line 236: 4 out of 5 cases were above 84%?

The 5 CSET cases had 84%, 84%, 84%, 74% and 83% surface RH (table 3, Sarkar et al. 2020).

Line 243: 'slightly lesser' is awkward

This line is now removed.

Line 252 and Fig 7: I see vertical profiles in Fig 7. I don't understand what the cases denoted by altitudes represent. Altitudes of what?

The figure 7 is now improved to include all the 22 cases instead of just 4 cases. The plots show the modeled outputs and the altitudes refer to the altitude at which the model parameters were computed.

Line 327: "independently evaluating" the modeled P3 cases is misleading. There are no validation observations at the surface.

The Brown, Meteor and BCO surface d-excess observations are used to evaluate the model, by running the model at the surface station observed RH. We have clarified this now in section 3.4.2.

Line 335-341: The differences between Salamalikis and this study for 2 mm drops seems quite large: 64 permil vs 27 permil for dD?

We found some printing mistakes in the Salamalikis et al. paper in some of their empirical values and a formula. Due to these issues, we have now referred to Graf et al. and Pruppacher and Klett work wherever required. Due to this, we have removed this section where we compare the Salamalikis results with ours.

3.4.1 subheading should include modeling like the 3.4.2 subheading

We have taken care of these during the re-editing.

Line 372: give range of d-excess values rather than the spread

The ranges for d-excess along with the dD and d18O are now given.

Line 389-390: this may be an overstatement

The revised model shows a good agreement between the modeled surface d-excess values and the surface observations for those P3 cases where RH at surface was above 75%. Four such P3 have modeled d-excess between 8-11 permil which falls in the observed d-excess range. Additionally, when the model is initialized with 85% for all the 22 P3 cases, then the range of modeled d-excess increase to the observed range. This is better shown in figure 10 histograms. Based on these, we have evaluated the accuracy of the model, explained in section 3.4.2.

Line 392: remind me how the P3 case surface RH is measured? From the drop sondes?

Yes, the RH at surface is obtained from the dropsondes.

line 412: Eqn 10 instead of 9, but the eqn doesn't show weight.

This paragraph has now been rewritten.

Line 416: The conclusion that evaporated water from rain drops doesn't influence the atmospheric vapor isotopic composition might not extend to other cases outside the tropics in drier air masses.

We agree. This conclusion may not be true for cases with stronger Fe or different microphysical conditions, as we have now clarified in the manuscript.

Fig 3: It would be more intuitive to stack the legend labels from highest altitude at the top decreasing toward the bottom.

The legends are ordered based on their time of sampling. We did not find any trend between the altitude of sampling and rain rates. Also, the cases do not represent the same cloud cell. Therefore, we keep the order based on their sampling time for ease of reference.

Fig 6: boxes are difficult to see in my printed version.

The size of the plot is now increased.

Fig 7: Is this modeling for the P3 case or Ron Brown case?

The modeling is done only for the P3 cases.

Fig 9: Is the red modeled or observed RSD at 700 m? While reading the description of this figure, it's difficult to see the features that are described in the text. The relationship between droplets at 700 m and the surface are not indicated. Would arrows help? Is the log-normal fit important or can that be removed? Given the log scale, it's difficult to identify important sizes like 700 and 900 micrometers.

We have removed this figure now since we do not need it to describe our story.

Fig 11: Edit delta symbols. What RH was this model run conducted?

This figure is also removed since the grid resolution for the new model is improved from 50 m to 1 m.

Fig 12: the stacked color bars do not print well. Separate histograms?

We have now used a line histogram that is easier to comprehend.

Fig A2 and others: I don't understand the "altitude for each case." Each case is plotted across all altitudes (0-700 m). For example, in panel d, 2 lines are shown labeled 1354 m with only 1%

difference in RH, but the RWCs are extremely different. If "case altitude" and RH aren't important, what is?

Yes, there are cases where for similar RH and sampling altitudes, the RWC is different. This is because measurements were made at the same altitudes for separate cloud systems. This gives us two different RSDs for the two different clouds measured at the same altitudes. For such cases, it is the difference in RSD that is responsible for the differences in RWC.

---

## Author Comment (AC2)

**Response to referee comments on "Sub-cloud rain evaporation from shallow convection in the north Atlantic winter trades" by M. Sarkar, A. Bailey, P. Blossey, S. d. Szoeke, D. Noone, E. Q. Melendez, M. Leandro and P. Chuang.**

We thank both the referees for their recommendations and comments on the manuscript. We believe these feedbacks have made our manuscript more comprehensive and informative. The referee comments are shown in blue and author answers in black.

Reply to Referee #2 comments:

Major changes have been incorporated throughout the manuscript since the last submission in terms of paragraph organization and story-telling. The introduction and conclusions are rewritten to make the paragraphs tighter and clearer. The broader implications of this work are also discussed in the introduction. The comparison with previous campaigns has been removed from the introduction and discussed more elaborately in the results and conclusions sections. Some figures have been removed, others improved, to better represent the theme of our work.

In addition to some unit corrections in the code, the vertical resolution is also improved. Instead of a fixed 50 m vertical resolution, a nominal step size of 1 m is used. In addition, an adaptive step size is employed to make the model stable and adaptive for very small droplets. The MATLAB code function is made available to the ACP for publication. The improvements have led to an increase in the overall rate of evaporation. This has increased the total column rain evaporation fluxes and the surface d-excess of rain.

This paper uses observations from the 2020 EUREC$^4$A field campaign in combination with a one-dimensional model that simulates the change in drop size and isotope composition of the drops for given initial conditions that are constrained by aircraft observations (raindrop size distribution, isotope composition of the drops). I very much enjoyed reading this nicely written paper. Below cloud evaporation is a strongly under-researched topic. Since it is one of the two components of precipitation efficiency (together with conversion efficiency in clouds), the lack of constraints on below cloud evaporation from observations has important consequences for our ability to correctly predict precipitation at the weather event as well as the climate timescale.

I have four major comments, listed below, as well as a short list of minor/technical comments below:

Major comments:

A)  It would be very valuable to point out the more general implications of this work already in the introduction (as well as at the end of the conclusion) e.g. trying to constrain precipitation efficiency with isotope observations would be one aspect to add to the now rather narrowly focused introduction. Of course, there are other aspects such as the impact of moisture recycling on mesoscale organization, which might be what the authors are more interested in. Right now the introduction reads like a nice summary of reference values of subcloud layer rain evaporation $F_e$ in terms of energy input into the subcloud layer, but these values would be much more interesting to compare with the author's results later on in the discussion.

The introduction has now been revised to discuss more general implications of this work which includes the rain evaporation flux contribution to BL stability and large-scale circulations, on the link of rain evaporation efficiency to cloud albedo and surface rain estimates, and the overall understanding the rain lifecycle. The Fe values from previous campaigns have now been removed from the introduction. They appear later in the results and conclusions sections.

B)  Given the motivation of the authors to investigate below cloud evaporation, because it represents a substantial energy and humidity input into the subcloud layer, I think that the results from Section 3.6 are very disappointing. Can this aspect be discussed in more details? It seems unlikely that such an important process would leave no distinguishable isotope signal in ambient vapour. Do these results imply that, even though below cloud evaporation strongly impacts the amount of precipitation that reaches the surface, for the subcloud layer moisture budget, it is a negligible process? Or is it only important, when integrated over larger spatio-temporal scales than a single precipitating cell?

Our results indicate that shallow rain evaporation with cloud base rain rates as low as the average P3 cases with ~1 mm/day could produce substantial evaporation fluxes and cooling rates for the sub-cloud layer. This could have BL stability implications depending on the vertical flux structure. This evaporation signature is easily measurable from the rain isotope ratios at the surface.

However, whether or not, these evaporation signals would be detectable by the vapor isotope analyzers depends on two conditions. Either the evaporated flux Fe needs to be high, or the evaporated vapor needs to accumulate in the sub-cloud layer for a sufficient time without advecting or diluting into the surrounding air. Sufficiency of either of the two cases, would ensure the detectability of the change in the vapor isotope ratios.

For example, considering the highest raining case during the P3 which has a maximum of 2 Wm-3 of Fe, if the evaporated vapor accumulates over 10 minutes, then the change in absolute humidity in the sub-cloud layer would be 0.5 g/m3. For an observed dDv of -71 permil and dDe of 5 permil, this will yield 2.4 permil of isotope change in dDv which is well-perceivable with the airborne isotope analyzers.

If Fe were smaller, then the accumulation time needed by the vapor to make a measurable change in dDv should be longer. It is also possible that in rain cells with more strong precipitation rates than the P3 cases, Fe would be higher. This might be the case for the Brown observations or the ATR observations made aboard the French ATR-42 (ATR) operated by SAFIRE. Both these platforms had rain rates more intense than during the P3. For such more intense rain cases, even over short time intervals, the Fe could be large enough to be detectable by the vapor isotope analyzers. However, ascertaining this is beyond the scope of our study.

How much does the authors' finding depend on the uncertainties of the aircraft and ship-based observations?

The uncertainty in the vapor dDv and d18Ov measurements from the aircraft is quite small (2 permil and 0.8 permil, respectively) for altitudes with higher water vapor concentration such as in the sub-cloud layer that we are interested in (figure 8 Bailey et al. 2023). The rain isotope ratio observed from the Brown have even smaller uncertainties of 0.8 permil and 0.2 permil for dDp and d18Op, respectively (table 1 in Bailey et al. 2023). These estimates are now also mentioned in the data sections of the manuscript (L104-118).

Also, can this aspect really be assessed with the model at hand given the assumption that vapour contributed by rain evaporation is neglected (L. 125).

The assumption of negligible contribution of vapor by rain evaporation in the model was confusing, and it has now been removed. Because the rain evaporation model is in steady-state, it is implied that the effect of vapor from evaporation on any future rain is ignored. The ambient vapor is obtained from the aircraft measurement close to the surface. This vapor is assumed to include the vapor from rain evaporation that has taken place already. Therefore, any rain evaporated vapor computed in the model is not further added to the background vapor.

C) There is very limited literature about below cloud evaporation effects, I agree, but I think there are a few studies from different settings with which the results in this paper can be compared to. For example, Aemisegger et al. 2015 GRL used a combination of numerical experiments and isotope observations to assess the importance of below cloud evaporation for a cold front passage. They found that over the whole frontal precipitation event neglecting below cloud evaporation leads to depletion biases of 20–40‰ in d2Hp and 5–10‰ in d18Op as well as to an increase of 74% in rainfall amount. This impact on total rainfall amount is very close to what the authors find in their study over the tropical North Atlantic. Also, in this paper, a substantial impact of below cloud evaporation on ambient vapour was found. How comes that in the winter trades the impact is so small? (Different region, different dynamics).

The major difference between the Aemisegger et al. 2015 case and the P3 case studies is the intensity of rain rates. The surface rain rates in the Aemisegger et al. study is 1-7.5 mm/hr which is substantially larger compared to the P3 cases. Higher rain rates could be due to higher Dg and sigma at cloud base that eventually reached the surface, and hence to higher Fe. Higher Fe cases can produce higher concentration of evaporated vapor over a given time, which could be detectable in the measured vapor isotope ratios as shown in the Aemisegger et al. study.

We suspect that because the rain rates during the P3 were sufficiently lower compared to the Aemisegger et al. study, the P3 Fe might also be smaller. Therefore, for the P3 cases the evaporated signal would be detectable if integrated over a longer time. Perhaps, the Aemisegger et al. study would be better compared with the ATR cases where rain rates are significantly higher than the P3.

D) The discussion of the impact of $F_e$ on stability is interesting but confusing. The statements at L. 274ff and in the conclusions (L. 432) are contradicting. Please clarify. I don't understand, based on which of their findings they draw these conclusions on stability. If $F_e$ has limited to no impact on subcloud layer temperature and specific humidity (see also major comment B) then how can stability be impacted?

The impact of Fe on stability has now been elaborated in the revised manuscript. Using the model results we define the top- or bottom- heaviness of the Fe profiles, and then see how they relate to the microphysical parameters. This is shown in figure 8b-d and discussed in section 3.3.3. The model results show that the top-heavier profiles are linked with smaller Dg and sigma (but not N0), and vice-versa. We also find that a low $RH_{sf}$ is also linked to more top-heavy profiles.

The possibility of the Fe vertical structure influencing the BL stability is based on previous studies done by like Srivastava 1985, Paluch and Lenschow 1991 and others, where rain evaporation closer to cloud base or surface has been linked with BL stability.

Additionally, it is important to note here that even if the evaporated vapor concentration is small compared to the ambient vapor concentration, it could still be energetically significant. This is because even for small vapor perturbations, FeT is still 10-350 Wm-2, with either top- or bottom- heavy vertical structure. Even if the vapor does not accumulate over a long period of time, the energy produced in that time instant could still potentially influence the local vertical circulation or contribute its moisture and energy to other rain systems in its vicinity. However, LES studies would be necessary to ascertain these effects.

Technical comments:

1) I think ACP titles are usually not capitalized and I would strongly encourage the authors to mention more specifically the type of precipitation events they are looking at: "Sub-cloud rain evaporation from shallow convection in the North Atlantic winter trades"

The title is now changed to "Sub-cloud rain evaporation from shallow convection in the North Atlantic winter trades".

2) The variables should be in italics except for abbreviations such as RH.

The necessary variables are now italicized.

3) L. 5 not sure I immediately understand, in which phase dD and d18O were measured and used in the model

The phase is now mentioned. (L5)

4) L. 9: 65% of what? mass, volume, event duration, number of events?

65% of mass. This is clarified now. (L7)

5) L. 17: is precipitation in shallow convection regimes really "ubiquitous"? I would have said it is rather sparse with low precipitation efficiencies compared to other cloud systems?

Thanks for pointing that out. Ubiquitous has been substituted by sporadic. (L18)

6) L. 48: "facilitate or hinder boundary layer stability" sounds a bit strange, how about "Does Fe reinforce or weaken the subcloud layer stability?"

The line is now rewritten as "Could Fe reinforce or weaken….". (L46)

7) L. 58-60: "This is because as rain evaporates…". I am not sure I can follow the implication that is formulated in this sentence. Vapour isotopes can be used independently to assess rain evaporation because rain evaporation leads to an enrichment of rain? This also joins my major point B above.

The sentences have been rewritten as: "In-situ measurements also provide stable isotope ratios of hydrogen and oxygen in water vapor, which can be used to independently assess rain evaporation. This is because as rain evaporates into the unsaturated sub-cloud layer, the isotopically light water transitions to the vapor phase more efficiently, causing the drops to become increasingly heavy (Salamalikis et al., 2016; Graf et al., 2019)". (L56-59)

8) L. 114: degree W formatting

Done.

9) L. 124: what is the implication of ignoring collision-coalescence for your results?

We assume that any collision-coalescence process between the altitude of aircraft leg and the cloud base is negligible. This essentially ignores any change in RSD from the altitude of sampling and cloud base. The RSD at the sampling altitude is then assumed to represent the RSD at cloud base. This assumption is drawn from the proximity of the microphysical parameters for the similar rain rates but measured at different altitudes (figure 5). The assumption also ties with a stratocumulus study in Wood (2005a) where rain rates remain constant in the lower 60% of the cloud.

10) L. 125: This last sentence leads to confusion about how you can assess the impact of below cloud rain-vapour interaction with the chosen 1D modelling approach.

The line was misleading and is now removed.

11) L. 134: could the variables be described one after the author, instead of the long list of variables and then a long list of descriptions? (would be easier for a reader like me to grasp).

Done.

12) L. 150: "." Formatting, should go to L. 149.

Done.

13) L. 161: all parameters from Graf 2017: this is a bit vague. Which exactly and is this a good choice given the large contrasts between a cold front in the midlatitudes and shallow convection in the tropics? Maybe a summarizing table in the Appendix would help.

In the modified code, the parameters are re-checked and modified wherever necessary. The details and description of all the parameters are given in section 2.2. The code is also now made available to ACP.

14) P. 5: what is the imp act of RWC estimates from P3 observations on the modelling? How do they compare to the ATR observations from which large statistics are available at the cloud base level. See Bony et al. 2022 ESSD.

The ATR estimates in figure 10 in Bony et al. 2022 is based on cloud drops (5-80 microns) and the median LWC is 0.05 g/m3 at cloud base. The P3 estimates in our work is for raindrops with diameters of 0.125-6 mm. Therefore, the comparison might not be appropriate. However, a quick check with ATR remote sensing files shows that ATR RWC from the BASTALIAS RWC product has 0.8 g/m3 maxima. This is higher than the highest RWC of 0.1 g/m3 from the P3 cases. The higher ATR RWC could have different influence on the rain evaporation than the P3 cases. But this is beyond the scope of this study.

15) L. 184: this seems to contradict the statement at L. 125. Wouldn't an integration over longer time intervals (precipitation events of 10-30 min) be necessary to assess this aspect?

We agree. In the revised manuscript, we have considered the time interval over which the change in delta_v could be measurable by the isotope analyzers.

16) L. 274ff: is this speculative, or based on some specific results? And what dominates for changing stability: the evaporative cooling or the moistening effect (which are counteracting each other)?

This paragraph has been clarified under its own subsection (section 3.3.3). We have used our model to quantify the relationship of the microphysical parameters on the vertical structure of Fe. Then we have used the results of previous studies like Paluch and Lenschow (1991), Srivastava (1985), Sandu et al. (2011) to discuss the implication of Fe profile on BL stability.

17) L. 285-294: I like this finding a lot. Very clearly explained!

Thanks. We have now replaced the figure with a scatter plot to make our estimates more quantitative. Figure 8b-d, section 3.3.3.

18) L. 357: Also this finding is very interesting! Maybe: "Consequently, the amount effect may not be appropriate for describing the impact of rain evaporation on the isotope composition of rain"? Did the authors' consider using the Graf et al. 2019 ACP deltadeltaD vs. delta-dexcess phase space for assessing the impact of different N0, cloud base RWC, RHsf etc.?

We have not used the deltaD vs delta-dexcess space in Graf et al. 2019 for this study, but this could be definitely useful in a more detailed study to investigate the impact of microphysical and thermodynamic parameters on the isotopic composition of rain.

19) L. 415: this suggests as a consequence that the impact of sub-cloud evaporation on stability is negligible too (see again my major point B).

This paragraph has been rewritten considering that the rain evaporated vapor concentration accumulated over time could have measurable impact on the observed isotope measurements.

20) L. 418: North Atlantic

Done.

21) Conclusions: could the authors point towards bigger implications and open up on further research that could be done e.g. for better constraining below cloud evaporation in models?

Done.

---

## Referee Report (RR1)

General comments:

I've reviewed the authors' revisions and response to both reviewers' comments. I find the revised manuscript much improved and nearly ready for publication.

Regarding value of the work to the field, I agree that the evaluation of the subcloud BL and microphysics is important for our understanding cloud-topped boundary layers. What remains in my view, is whether this study shows that rain isotope observations constrain the rain evaporation estimates independent of other estimation methods. If the authors have quantitative support of that question, adding it would increase the impact on the field. It is a very nice contribution to make this model available to the community.

The new model results improve agreement with the observations on the Ron Brown ship. I agree that d-excess is most sensitive to evaporation and should be the metric of analysis. The previous presentation made it hard to see the full picture of the isotopic evolution though the BL. The isotopic presentation is improved. Figure 10: It is my understanding that the authors consider the modeled rain at 85% RHsrf (magenta line) is in good agreement with the observations from the Brown (red line). I don't see any of the model results in Fig 10 that get above 10 permil, but Fig 11 model results extend to 11.5 permil. Perhaps it is my ability to read the figure line colors, but if there is an inconsistency, please fix.

The linear relation between REF and dp features prominently in the key findings of this paper, yet the figure is in Supplemental with no statistical metrics given. Please give statistical analysis and uncertainty at minimum. There are some values that could use some discussion in the figure caption or the main text if warranted. Values of REF=1.0 indicate the rain evaporates completely therefore d_p,sf is NA. Likewise rain isotope observations where d-excess increases instead of decreases are curious given the conventional assumption that d-excess decreases with evaporation influence.

Specific comments:

Page 7, line 205: Can you be more specific about what an appropriate integration time would be?
Section 3.2: When discussing correlations in Figure, provide quantitative correlation and statistical significance. The correlation described in Fig 5d may not be significant.
Page 12, line 362: "Most of the P3 cases with higher $D_g$ and $\sigma$ also have higher $N_0$." Is this expected or unusual? It seems important for the generality of the findings.
Page 14, line 402: "These ranges correspond well to the value show across other platforms…" What other values? The 3 ship platforms are the only rain measurements mentioned so far.
Page 14, line 406: "…with a high RHsfc of 86% is around 10 permil that matches the brown…" awkward wording.
Conclusions: Conceptual question: To what extent does the thermodynamic state of the BL control rain evaporation versus rain evaporation control the thermodynamic state of the BL?

Page 16, line 480: The linear relation between REF and dp features prominently in the key findings of this paper, yet the figure is in Supplemental with no statistical metrics given. Please give statistical analysis and uncertainty at minimum.

Fig 1: Does 'rain flux' is the same a 'rain rate' right? It's not clear to me what the profile in the BL represents. It seems labeled as the instantaneous rain rate, but that would have to highest at cloud base in both small droplet and large droplet cases right? Does blue indicate larger rain rates and red smaller? Does the x-axis also indicate the rain rate? If so, the partial evap case should end at different relative location than the complete evaporation rain rate, which would end at zero. Do the rain evaporation arrows indicate moisture recycling within those altitudes? Maybe the profile is the rain evaporation rate? It might be helpful to label this figure with the top/bottom heavy classification used later in the manuscript.

Fig 2: Label color bar as altitude (m).

Fig 4: Give units of color bar. Are they really plotted as contour lines? Or rather shaded?

Fig 5: Give statistical metrics of relationships/no relationships.

Fig A1: Give correlation statistics.

Fig 7: The cases where dDp decreases with altitude are curious. I think d is used throughout the manuscript instead of dxs. Please be consistent with the label in this figure.

Fig 9: Breakpoints suggest this figure may have been generated using model with 50 m altitude steps (previous version).

Figs 10-11: See previous comment.

---

## Author Response (AR2)

**Response to referee comments on "Sub-cloud rain evaporation in the North Atlantic winter trades derived by pairing isotopic data with a bin-resolved microphysical model" by M. Sarkar, A. Bailey, P. Blossey, S. d. Szoeke, D. Noone, E. Q. Melendez, M. Leandro and P. Chuang.**

We thank both the referees for their recommendations and suggestions on the manuscript. These feedbacks have helped us refine our results and made the overall presentation better. The referee comments are shown in blue and author answers in black.

Reply to Referee #1 comments:

This study attempts to understand the sub-cloud rain evaporation via the simulated rain evaporation flux and water vapor isotope ratios using a one-dimensional model. I think overall the authors well address the comments brought by the reviewers in the first-round review. I recommend it for publication with some minor revisions, which are listed as below.

Title: The current generally looks good, but given the primary method and the significant portion of the study is related to a one-dimension model simulations, I feel that the title should be improved if the model was incorporated, which was also mentioned by anther reviewer in the first-round review.

We have changed the title to include the model and observation use:

*"Sub-cloud rain evaporation in the north Atlantic winter trades derived by pairing isotopic data with a bin-resolved microphysical model".*

Line 4: Since here the authors mentioned that the change in temperature was also simulated, I expected there would be some discussions regarding temperature-related results later but actually not.

The mean ambient temperature at cloud base remains fairly consistent between 292-293 K over all the cases studied here. Following this, we did not find any interesting contrasts in the rain evaporation characteristics due to temperature differences.

Line 11: 'between cloud base and the surface, as compared to a 'bottom-heavy' profile' -> 'between cloud base and the surface than a 'bottom-heavy' profile'.

Done.

Line 13-14: It's better to move model performance evaluation to earlier place in the abstract, e.g., before reporting essential results. How about the model performance of other parameters?

As you suggested, we have moved the model evaluation earlier in the abstract now.

Lines 148-166: it is not necessary to separate each parameter in each line, which can save space of the body text.
We tried writing the parameters within a paragraph, but that was hard to read. In this format, we feel the clarity was more.

Lines 355-356: From Fig. A2, it seems that the REF dramatically decreases with N0, and the decreasing slope is not small compared to the case in Dg-REF or sigma-REF relations. Be cautious to draw a conclusion as "The influence of N0 on REF is smaller compared to Dg and σ (Figure A2d). This is because when REF is expanded in terms of N0, Dg and σ, N0 appears in the numerator and denominator and almost cancels out".
To emphasize on the stronger relationship between Dg and sigma with REF compared to N0, the correlation coefficients (r) has been added to the plot. The r for N0-REF is -0.3 compared to -0.7 and -0.8 for Dg and sigma, respectively.

Line 360: add "and" before "N0 appears in the ....".
We have added 'the' before 'N0 appears in the ...' which seemed more appropriate.

Line 370: "figure 9"-> "Figure 9".
Done.

Line 432: "equation 11"->"Equation 11".
Done.

Line 484: Replace "But" with "In contrast".
Done.

Code availability: Just specify where the code could be accessed. I agree with another reviewer in the first-round review that "it could be useful for providing Monte-Carlo estimates of surface rain isotopic composition in future studies."
The code is submitted to the ACP and should be available to readers upon the manuscript publication.

Fig. 4. The color bar label is missing.
Added.

Fig. 5: Panel number annotation is missing. Also delete "Only the average Dg and σ over five CSET cases were available and are shown by single dots in b-c)". Also add a legend to denote what the colors of circles stand for.

Panel numbers are added along with some statistical parameters. A legend is also added. We have retained the statement "Only the average Dg….", since this will clarify why only one data point for Dg and sigma is provided as opposed to five data points for No and RH_surface.

The figure has been revised now.

Legend is added. The line "The dashed …" is removed. The slope information is revised by adding the fit equation.

Reply to Referee #2 comments:

General comments:

I've reviewed the authors' revisions and response to both reviewers' comments. I find the revised manuscript much improved and nearly ready for publication.

Regarding value of the work to the field, I agree that the evaluation of the subcloud BL and microphysics is important for our understanding cloud-topped boundary layers. What remains in my view, is whether this study shows that rain isotope observations constrain the rain evaporation estimates independent of other estimation methods. If the authors have quantitative support of that question, adding it would increase the impact on the field. It is a very nice contribution to make this model available to the community.

The rain evaporation estimation is calculated directly from the isotope-initialized microphysics-resolved model. The proportionality between the difference of d-excess between the cloud base and surface and the rain evaporated fraction supports that the isotope observations could be a reliable method to study rain evaporation. We have now added correlation coefficients in the necessary plots to emphasize on this.

This RSD-isotope combination method works independent to other methods such as radar-based rain evaporation. A future study could be formulated to compare the rain evaporation estimated from the two methods. We have now added a paragraph in the conclusion describing this:

*"In general, our isotope-initialized microphysics-resolved model performs reliably well in characterizing the sub-cloud rain evaporation in the shallow rain regime sampled during the ATOMIC/EUREC$^4$A campaign. This model also only requires in-situ microphysical and rain isotope observations and is independent of any remotely-sensed rain observations. However, a comparison between rain evaporation evaluated from remote-sensing platforms (e.g., mm-wavelength radars) and our in-situ-based model could be useful for error analysis."*

The new model results improve agreement with the observations on the Ron Brown ship. I agree that d-excess is most sensitive to evaporation and should be the metric of analysis. The previous presentation made it hard to see the full picture of the isotopic evolution though the BL. The isotopic presentation is improved. Figure 10: It is my understanding that the authors consider the modeled rain at 85% RHsrf (magenta line) is in good agreement with the observations from the Brown (red line). I don't see any of the model results in Fig 10 that get above 10 permil, but Fig 11 model results extend to 11.5 permil. Perhaps it is my ability to read the figure line colors, but if there is an inconsistency, please fix.

We realize that the extent of the P3 domain in the contour plot (now removed) not matching the P3 modeled values seemed confusing. The P3 domain box in the contour was not intended to represent the d_p. It was only included to provide the Dg and sigma ranges during the P3 and compare it with the CSET ranges. Since the 22 P3 cases have different RH_surface (66-87%), and the contours in figure 11 were run at constant RH of 70% and 85%, the d_p over the P3 box did not match the P3 modeled values.

We have now removed the contour and shown the effects of RH, Dg and sigma on d-excess in terms of histogram lines in the new Figure 11b. We have run two run at a) RH_surface=85% and Dg=0.5 mm, sigma (1-2.5) from P3 cases (yellow line) and, b) RH_surface=85%, Dg=0.5 mm, sigma=3 (purple line). The d_p for these two runs is higher than the default P3 cases where Dg, sigma and RH were generally lower. The higher modeled d_p ranges for the two new model runs are also closer to the Brown observations. Along with evaluating the model with the Brown observations, these experiments also show that both thermodynamics and microphysics contribute to the higher d_p.

Since this plot is crucial to show our model evaluation with the observations, we hope that the interpretation is more comprehensible now.

The linear relation between REF and dp features prominently in the key findings of this paper, yet the figure is in Supplemental with no statistical metrics given. Please give statistical analysis and uncertainty at minimum. There are some values that could use some discussion in the figure caption or the main text if warranted. Values of REF=1.0 indicate the rain evaporates completely therefore d_p,sf is NA. Likewise rain isotope observations where d-excess increases instead of decreases are curious given the conventional assumption that d-excess decreases with evaporation influence.

The aforementioned figure is now added to the main manuscript (figure 8) with a linear fitting equation, correlation coefficient and the p-value.

The cases with very high REF tend to but are always smaller than 1. The one remaining case (1 out of the 22) on 4th February where rain completely evaporated (REF=1) is not included in the plot since its d_sf is NA (just as you explain). A note has been made about this in the caption.

The cases with 1-d_sf/d_cb greater than 1 are due to d_sf being negative (see figure 7i for reference). This makes 1-d_sf/d_cb greater than 1. This note is also added now in the caption.

Specific comments:

Page 7, line 205: Can you be more specific about what an appropriate integration time would be?

The time series of the surface rain measurements at the Brown station last around 10 minutes. There are longer showers as well but the signal is very patchy. This gives us confidence to use an approximate length scale of 15 minutes to compute the accumulated rain evaporated vapor in section 3.5.

Section 3.2: When discussing correlations in Figure, provide quantitative correlation and statistical significance. The correlation described in Fig 5d may not be significant.
Both the correlation coefficient and p-value are now added to describe the plots. The correlation between RH_surface and rain rate are indeed weaker than the other three microphysical variables.

Page 12, line 362: "Most of the P3 cases with higher Dg and s also have higher N0." Is this expected or unusual? It seems important for the generality of the findings.

A general relationship between Dg, sigma and N0 is unclear especially considering the variability in rain rates and the microphysical parameters. Geoffroy et al. (2014) showed that rain at a mature stage is characterized by broader RSDs centered at larger drop sizes, and with slightly decreased but still high N0. This is closer to our finding. They further find that near the surface sigma gets narrower, and N0 is high if rain water content is high, and vice-versa. However, they note that the study is limited by a low number of raindrops and instrumental biases.
In another previous study, Feingold and Levin (1986) found that for low rain rates (<5 mm/hr), Dg, sigma and N0 vary considerably. But as rain rates get higher, sigma is remarkably constant while N0 and Dg are increasing. Despite these observed trends, other variations in the Dg, sigma, N0 correlation were also observed. Overall, the correlation of Dg, sigma and N0 seems to be dependent on the life stage of the rain, rain water content and the microphysical processes involved.

Page 14, line 402: "These ranges correspond well to the value show across other platforms…" What other values? The 3 ship platforms are the only rain measurements mentioned so far.
Other than the 3 ship platforms mentioned in the manuscript, the precipitation samples were also collected by the German research vessel *Merian* and French research vessel *Atlante*. Their ranges are provided in Bailey et al. 2023 Figure 11.

Page 14, line 406: "…with a high RHsfc of 86% is around 10 permil that matches the brown…" awkward wording.
The paragraphs in the section have been revised to make the interpretation more straightforward.

Conclusions:
Conceptual question: To what extent does the thermodynamic state of the BL control rain evaporation versus rain evaporation control the thermodynamic state of the BL?

The BL thermodynamic state and the rain evaporation are interdependent. The thermodynamic state at which a rain shower occurs determines the amount and distribution of the rain evaporation. Conversely, depending on the distribution, intensity and duration of the rain evaporation, the thermodynamic properties around the rain shaft will also change.

Page 16, line 480: The linear relation between REF and dp features prominently in the key findings of this paper, yet the figure is in Supplemental with no statistical metrics given. Please give statistical analysis and uncertainty at minimum.
Done.

Fig 1: Does 'rain flux' is the same a 'rain rate' right? It's not clear to me what the profile in the BL represents. It seems labeled as the instantaneous rain rate, but that would have to highest at cloud base in both small droplet and large droplet cases right? Does blue indicate larger rain rates and red smaller? Does the x-axis also indicate the rain rate? If so, the partial evap case should end at different relative location than the complete evaporation rain rate, which would end at zero. Do the rain evaporation arrows indicate moisture recycling within those altitudes? Maybe the profile is the rain evaporation rate? It might be helpful to label this figure with the top/bottom heavy classification used later in the manuscript.

Thanks for pointing these out. The schematic has been retouched to address these issues.
A bulk rain evaporation flux of 28 W/m2 that is produced from the evaporation of 1 mm/day of rain is shown in the schematic. The confusion was due to the '=' sign and has now been replaced by '→' sign. The x-axis of the vertical lines is depictive of rain evaporation fluxes and are now modified for the complete and partially evaporating cases. The shades were removed as it was unnecessary. The top- and bottom- heaviness of the profiles are now labeled. The lines are not intended to be scaled. Isotope information has also now been added.

Fig 2: Label color bar as altitude (m).
Done.

Fig 4: Give units of color bar. Are they really plotted as contour lines? Or rather shaded?
Done. The contours are shaded and the shade is represented by the color bar.

Fig 5: Give statistical metrics of relationships/no relationships.
Done.

Fig A1: Give correlation statistics.
Done.

Fig 7: The cases where dDp decreases with altitude are curious. I think d is used throughout the manuscript instead of dxs. Please be consistent with the label in this figure.
The decreasing dDp with altitudes are for cases with a combination of high RH and relatively more enriched rain at cloud base. Additionally, this issue only happens for smaller drops in the RSD. We suspect that the higher RH subjected to these raindrops leads to stronger equilibration

signals as compared to evaporation signals. Since the rain in these cases are also more enriched, it could be losing the heavier isotopologues due to the exchange process and getting depleted.

Fig 9: Breakpoints suggest this figure may have been generated using model with 50 m altitude steps (previous version).
The breakpoints in figure 9 are due to the complete evaporation of smaller drops present in the RSD at a vertical level, producing the faint spikes in vertical Fe profiles. Similar breakpoints are also shown in figure 7f.

Figs 10-11: See previous comment.
Addressed.